# A compressive hyperspectral video imaging system using a single-pixel detector

Yibo Xu[1] ✉, Liyang Lu[2], Vishwanath Saragadam [3] & Kevin F. Kelly[3]

Capturing fine spatial, spectral, and temporal information of the scene is highly desirable in many applications. However, recording data of such high dimensionality requires significant transmission bandwidth. Current computational imaging methods can partially address this challenge but are still limited in reducing input data throughput. In this paper, we report a video-rate hyperspectral imager based on a single-pixel photodetector which can achieve high-throughput hyperspectral video recording at a low bandwidth. We leverage the insight that 4-dimensional (4D) hyperspectral videos are considerably more compressible than 2D grayscale images. We propose a joint spatial-spectral capturing scheme encoding the scene into highly compressed measurements and obtaining temporal correlation at the same time. Furthermore, we propose a reconstruction method relying on a signal sparsity model in 4D space and a deep learning reconstruction approach greatly accelerating reconstruction. We demonstrate reconstruction of 128 × 128 hyperspectral images with 64 spectral bands at more than 4 frames per second offering a 900× data throughput compared to conventional imaging, which we believe is a first-of-its kind of a single-pixel-based hyperspectral imager.

Hyperspectral video imaging captures 4-dimensional (4D) information of the scene containing 2D spatial, 1D spectral, and 1D temporal information represented by $X(x,y,\lambda,t)$. High resolution hyperspectral video imaging has become highly desirable in many applications for studying dynamic optical phenomena with complicated spectral information both in microscopic and macroscopic systems, such as biological fluorescence imaging, remote sensing, surveillance, and autonomous driving, etc. However, directly recording data of such high dimensionality requires large storage and significant transmission bandwidth. It may lead to significant power consumption, memory footprint, and time costs, imposing extreme pressure on imaging systems especially when storage and transmission of the data is critical, i.e., on satellites and rovers. On the other hand, it is well-known that the contents in adjacent frames of a video are highly correlated, and the image slices from nearby wavelength bands are very similar. As a result, 4D hyperspectral video data have higher inherent redundancy and far more compressible than 2D images, offering possibility to be

sampled and reconstructed at very high compression ratio or very low input data throughput.

Compressive sensing (CS)[1–3] is an effective solution to deal with the dilemma of limited bandwidth in hardware for high-throughput data acquisition. CS is a mathematical framework for efficient signal acquisition and robust recovery. Exploiting the inherent structure or prior of signals, CS enables us to stably reconstruct a signal sampled below the Nyquist sampling rate. CS allows for reducing the costs associated with sampling, transmission bandwidth and storage, especially when handling high-dimensional signals. Among the numerous applications inspired by CS, single-pixel imaging[4] (SPI) is a computational imaging technique that acquires coded projections of a scene using a photodetector without spatial resolution and computationally reconstructs the scene from the coded compressed measurements. In contrast to full-frame sensors having millions of pixel elements, the single-pixel detector design allows for high signal-to-noise ratio (SNR) and low-cost development of high-throughput cameras using exotic

[1]Beijing Engineering Research Center of Mixed Reality and Advanced Display, School of Optics and Photonics, Beijing Institute of Technology, Beijing, China. [2]Google Inc., 601 N. 34th Street, Seattle, WA 98103, USA. [3]Department of Electrical and Computer Engineering, Rice University, 6100 Main St, Houston, TX 77005, USA. ✉e-mail: ybxu2013@126.com

imaging modalities or beyond visible wavebands where conventional pixelated detectors are either too expensive, cumbersome or unavailable[5]. The SPI techniques have been demonstrated for a wide variety of applications, including infrared imaging[6,7], terahertz imaging[8], low light imaging[4,9], hyperspectral imaging[10–15], video imaging[16,17], and computer vision-related tasks[18–21], etc.

Reconstructing the signal from compressed measurements is an ill-posed problem in general, since the system of equations of measurements is under-determined. To solve the CS reconstruction problem, many optimization-based algorithms have been proposed[22–28]. Such algorithms generally use signal sparsity as prior knowledge to regularize the ill-posed inverse problem. The signal structure model or prior used greatly affects the reconstruction quality. For CS video reconstruction, the joint sparsity of signal in two spatial and the temporal dimensions such as 3DTV algorithm has been demonstrated[17,29]. The optimization-based approach enjoys the benefits of the flexibility of handling images of different sizes, compression ratios (CRs), and various types of signal priors, etc. However, optimization algorithms suffer from long computation time due to its iterative nature. Recently, deep learning methods for CS reconstruction flourished and have shown the potential to improve both the reconstruction quality and speed[30–34].

Hyperspectral imaging aims to obtain the spectrum associated with each pixel in the image of a scene in many narrow wavelength ranges[35]. Conventional scanning-based hyperspectral imaging methods are greatly speed-limited[36,37]. Compressive spectral imaging methods developed in recent years such as coded aperture snapshot spectral imaging (CASSI)[38], dual-coded compressive hyperspectral imaging (DCSI)[39], spatial-spectral encoded compressive hyperspectral imaging (SSCSI)[40], and various other spectral multiplexing mechanisms[32,41–43] perform spectral multiplexing to reduce the 3D data cube to the 2D array sensors. However, the compression in the measurements only happens in the spectral dimension, leading to limited compression ratio or limited capability in reducing input data throughput. Also, the 2D array detector used can be prohibitively expensive if not unavailable for wavelengths beyond the visible wavebands. Yako et al.[44] presented a video-rate snapshot hyperspectral imager where a CMOS-compatible random array of Fabry-Pérot filters is placed above a conventional 2D image sensor for pixel-wise spectral encoding. Reconstruction algorithms based on CS theory are used. Experiments demonstrated hyperspectral video reconstruction of spatial resolution of 480 ×640 pixels with 20 spectral bands at 32 frames per second (fps). In the sampling process, however, the signal encoding and compression happens only in the spectral dimension and not in the spatial dimension. The compression ratio is 20:1, thus having a 20x data throughput compared to conventional methods. Saragadam et al.[45] presented an adaptive hyperspectral imaging system which optically implements the so-called Krylov subspace method. Based on the low-rank assumption of the underlying image, it directly captures the dominant singular vectors of the hyperspectral image to compute its low-rank approximation. It allows hyperspectral imaging with 560 ×550 spatial pixels and 256 bands over visible wavebands. Besides being a multi-frame technique based on a 2D array sensor, the compression ratio in the experiments is less than 10:1. Saragadam et al.[46] introduced a video-rate hyperspectral imager by fusing the RGB image of the scene with the spectra sampled from a sparse set of spatial locations, utilizing scene-adaptive spatial sampling. The system allows hyperspectral imaging of 600 × 900 pixels with 30 spectral bands at a frame rate of 18 fps. The approach relies on assumptions of the image property precluding imaging certain types of scenes. Based on one RGB 2D sensor and one grayscale 2D sensor, the compression ratio in sampling is smaller than the number of spectral bands. Soldevila et al.[47,48]reported a time-resolved multispectral camera for fluorescence imaging based on a CMOS camera, a time-resolved bucket detector and a spectrometer coupled with a detector array. It produced a hypercube of 512 × 512 pixels with 16 spectral bands for 256 time-resolved frames while sampling 0.03% of all the reconstructed voxels. However, the system functions such that each sensor acquires a heavily downsampled version of the data which are upsampled and fused to produce the full-resolution data. It does not utilize any signal prior nor compressive sensing theory. Temporal information is obtained only from the time-resolved bucket sensor which reconstructs 256 low-resolution grayscale frames. The mechanism is designed for time-resolved fluorescence imaging and does not apply to normal motion scenes. Also, with heavy downsampling in each dimension, the reconstruction accuracy would be very low for normal spectral imaging.

Single-pixel approaches have been proposed for spectral imaging to avoid the limitations brought by array detectors[11–13,47,49]. Some of them use CS technique to reduce the amount of measurements, but due to the imaging system designs, such as the use of spinning wheels for spectral modulation[11,12], or fully mapping the multispectral data on to the digital-micromirror device (DMD)[13], and the two-stage spatial-spectral modulation models, e.g. using the same set of spectral modulations within every spatial modulation[49], the resolution and compression ratio of these systems are limited. Hahamovich et al.[48] developed an approach for rapid single-pixel imaging which uses a fast-spinning mask coded with cyclic sensing patterns achieving a spatial pattern modulation rate of up to 2.4 MHz. Experiments demonstrated reconstruction of grayscale images of spatial resolution of 101 × 103 pixels at 72 fps. The system utilizes 100% sampling with no compression in data acquisition for reconstruction. *Kilcullen* et al.[50] developed an SPI system accelerated via swept aggregate patterns that combines a DMD with laser scanning hardware to achieve pattern projection rates of up to 14.1 MHz and frame sizes of up to 101 × 103 pixels for real-time grayscale video imaging up to 100 fps. The highest compression ratio demonstrated is 4:1 in this system. Although achieving a high frame rate, these approaches increase frame rate by boosting spatial pattern projection rates, which directly leads to extremely short integration period for each measurement resulting in noisy measurements and very low SNR in the recovered grayscale image. Gutiérrez-Zaballa, et al.[51] presented a method for on-chip hyperspectral image segmentation through deep learning to accurately classify and segment objects to improve scene understanding in autonomous driving. The input to the neural network is the full 3D spectral image obtained from a 25-band mosaic spectral filter camera. Martins et al.[52] described an ecosystem for open-source hyperspectral "single-pixel" imaging, where hypercubes of 128 × 128 spatial pixels with 2048 spectral bands are acquired in 12 seconds. It also presents a database of more than 150 hypercubes. Note that the system relies on a DMD for spatial modulation and uses a commercial spectrometer as the detector which has a 2D array sensor. The spectral resolution of the hypercube is given directly by the spectral resolution of the spectrometer. Therefore, the term "single-pixel imaging" used in their paper means differently than that in our paper which refers to employing a sensor that actually consists of a single pixel photodetector. The spatial encoding scheme is based on subsampling the structured Hadamard matrix with the highest compression ratio of 32:1 demonstrated, and, without employing any compressive sensing method, shows relatively low reconstruction quality. No spectral compression or temporal correlation is involved. Bian et al.[11] proposed a multispectral imaging system that takes 60 seconds to image a 64×64×10 data cube. The system proposed by Li et al.[13] only captures 8 spectral channels of 128 × 128 images in 0.5 second. To minimize the number of measurements needed while still obtaining a high-quality high-resolution reconstruction, the inherent structure or the joint sparsity of 4D hyperspectral data needs to be fully exploited in both the sensing and the reconstruction stage.

In this paper, we report a hyperspectral video imaging system based on a single-pixel detector named the "Single-Doxel Imager" (SDI)

that can achieve high-throughput hyperspectral video recording at a low bandwidth. Here, a "doxel" is defined as a dynamic voxel or a pixel in the 4D space. We propose the framework for high-throughput hyperspectral video acquisition and recovery with the SDI. Specifically, we propose a CS-based sensing scheme using specially designed joint spatial-spectral modulation patterns displayed by a spatial light modulator, which encodes the scene into a highly compressed sequence of measurements, obtaining temporal correlation from optical flow at the same time. An optimization-based reconstruction algorithm is proposed that simultaneously exploits the high sparsity in 4D space. Experiments with this prototype SDI system demonstrate recovery of high quality 128 × 128 hyperspectral video data with 64 spectral bands at the speed of 4.3 frames/second, achieving compression ratio of 900:1 and offering a 900× data throughput compared to conventional imaging. To the best of our knowledge, this is the first time that hyperspectral video can be acquired at this compression ratio with a single-pixel detector. Furthermore, we develop a deep learning reconstruction approach for fast recovery, with the 4D temporal-spatial-spectral correlation in the signal embodied and incorporated in the neural network design. Trained on simulation data and tested on both simulation and experimental data, this approach demonstrates the feasibility of a fast reconstruction solution and promising reconstruction quality.

## Results

### Optics and hardware

The optical design of the SDI system is shown in Fig. 1a. A real image of the scene is formed through the relay lenses on the surface of a DMD (TI DLP Discovery 4100 with DLP9500 DMD chip) serving as a spatial light modulator through imaging lenses. The 2D spatial pattern displayed on the left part of the DMD, highlighted by a red square, spatially modulates the image on it and reflects the modulated light signal towards a light-collecting lens which focuses the spatially modulated image on a slit. The modulated light signal is the sum of the light reflected by the micromirrors at the "on" state on the DMD. The focused light signal on the slit is reflected toward the DMD with a pair of 45° mirrors. A diffraction grating then spreads the light signal spectrally and a focusing lens projects the spectrum onto the right side of the DMD. The slit is used to restrict light into the shape of a narrow rectangle. The focusing lens forms many copies of the image of the slit on the DMD each with a different wavelength due to the diffraction grating. On the right side of the DMD with the spectrally dispersed signal, a 1D barcode-like spectral modulation pattern is displayed as highlighted in a blue rectangle. The micromirrors at the "on" state inside the spectral pattern reflect the light towards the detector, while the ones at the "off" state discard the light on them. Finally, the spatially and spectrally modulated light signal is focused by a lens onto the single-element detector (Hamamatsu H9306) for the measurement. Details of SDI system setup and calibration are described in Section 1 of Supplementary Information.

### CS-based Joint Spatial-Spectral Encoding

During measurement process, the DMD displays a series of joint spatial-spectral patterns. Each pattern is composed of a 2D spatial

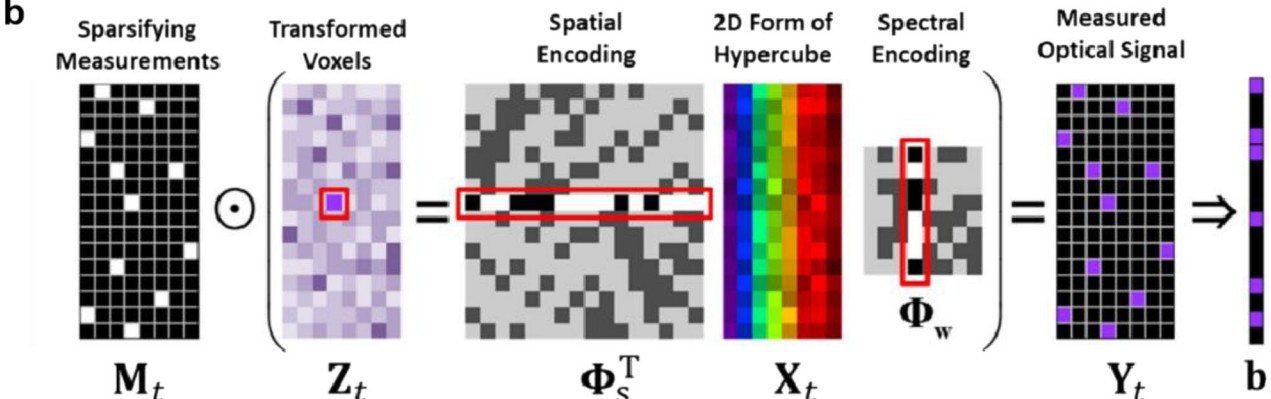

**Fig. 1 | Hardware design and sensing scheme of the Single-Doxel Imager. a** Schematic diagram of the Single-Doxel Imager system hardware design, **b** The measurements of one hyperspectral frame represented in a set of matrix multiplication.

pattern and a 1D spectral pattern displayed sided by side on a single DMD, as shown in Fig. 1a. One measurement is taken for every spatial-spectral pattern. This process can be modeled as a linear equation system illustrated in Fig. 1b.

As shown in Fig. 1b, one frame of hyperspectral video at time $t$ is denoted by a $N \times K$ matrix $X_t$ representing spatial resolution of $\sqrt{N} \times \sqrt{N}$ with $K$ spectral bands, where each column is a vectorized 2D image at one spectral band. The spatial modulation by the 2D spatial pattern on the DMD is equivalent to left multiplying $X_t$ with a $1 \times N$ row vector highlighted by a red rectangle, which is the vectorized 2D spatial pattern. The spectral modulation by the 1D spectral pattern on the DMD is equivalent to right multiplying $X_t$ with a $K \times 1$ column vector highlighted by a red rectangle. The product of the $1 \times N$ spatial modulation vector, the $N \times K$ data matrix $X_t$ and the $K \times 1$ spectral modulation vector is a value proportional to the light signal recorded by the detector, which is stored as one element in the transformed domain matrix $Z_t$ highlighted by a red square.

A complete set of measurements without compression involves using every combination of $N$ spatial patterns and $K$ spectral patterns from the full-rank $N \times N$ spatial modulation matrix $\Phi_S$ and $K \times K$ spectral modulation matrix $\Phi_W$, respectively. The complete measurement results compose the $N \times K$ matrix $Z_t$, containing all the information from the data $X_t$. For a $128 \times 128 \times 64$ hyperspectral frame, a complete set of measurements requires over 1 million modulations and data points, which is impractical and unnecessary for hyperspectral video imaging. With CS theory, we sparsely select the spatial and spectral pattern pairs used for the actual measurement, so only a small portion of the entries in $Z_t$ are filled with the measurement results, while all other values are kept unknown. The element-wise multiplication of a sparse 0-1 matrix $M_t$ with $Z_t$ in Fig. 1b represents this compressive subsampling process. The resulting matrix, $Y_t$, is the measurement data obtained from SDI system, with which the original signal $X_t$ can be reconstructed. As shown in Eq. (1), with the proper definitions, the measurements $b_t$ of one hyperspectral frame at time t denoted by $u_t$ can be expressed by a simplified linear equation. Here, $\otimes$ represents the Kronecker product.

$$b_t = vec(Y_t) \qquad R_t = diagvec((M_t))$$
$$u_t = vec(X_t) \qquad \Phi = (\Phi_w \otimes \Phi_s)^\mathsf{T} \qquad (1)$$
$$b_t = R_t\, \Phi\, u_t$$

During sampling of multiple hyperspectral frames, a longer DMD pattern sequence is used and a series of single pixel measurements are acquired over time. Along this train of measurements, we use the so-called "data window" to select a consecutive block of measurements within the window to define and recover each video frame. The length of the data window refers to the number of measurements within the window. We can "slide" the data window along the train of measurements to select the measurements for recovering each frame. We define the "sliding stride" as the number of measurements which the data window slides over going from one frame to the next. In principle, the lengths of the data window and the sliding stride for each frame can be freely chosen, depending on the needs for a particular scene. Equation (2) constructs the overall linear model for the hyperspectral video imaging process by the SDI, combining the matrices and data vectors of different frames. Here, $u$ is the data vector containing the unknown 4D hyperspectral video, $S$ is the temporal-spatial-spectral modulation matrix, $R$ is the compressive subsampling matrix, and $b$ is the compressive measurement vector containing the outputs from the SDI. The ratio between the amount of unknown data (the length of vector $u$) and the number of measurements (the number of 1s in matrix R) is called compression

ratio (CR).

$$
b = \begin{bmatrix} b_{t=1} \\ b_{t=2} \\ \vdots \\ b_{t=T} \end{bmatrix}
R = \begin{bmatrix} R_{t=1} 0 \cdots 0 \\ 0 R_{t=2} \cdots 0 \\ \vdots \vdots \ddots \vdots \\ 0 0 \cdots R_{t=T} \end{bmatrix}
$$
$$
u = \begin{bmatrix} u_{t=1} \\ u_{t=2} \\ \vdots \\ u_{t=T} \end{bmatrix}
S = \begin{bmatrix} \Phi 0 \cdots 0 \\ 0 \Phi \cdots 0 \\ \vdots \vdots \ddots \vdots \\ 0 0 \cdots \Phi \end{bmatrix} \qquad (2)
$$
$$ b = RSu $$

The modulation patterns used for sensing have pseudo-random structures making them suitable for CS. For the experiments in this paper, the spatial modulation patterns in $\Phi_S$ are from the "structured random" STOne pattern sequence[17]. The STOne patterns enable multi-resolution image reconstruction. Details of the properties of STOne patterns are presented in Section 2 in Supplementary Information. The spectral patterns in $\Phi_W$ are pseudo-randomly permuted Walsh-Hadamard patterns which are used to provide the randomness needed for CS-based reconstruction. Both the spatial and spectral patterns are binary patterns with {+1, −1} entries that can be easily implemented on the DMD utilizing the so-called complementary patterns (see *"Methods"*). The complementary patterns are also used to obtain pure spatial modulation from the raw measurements for calculating grayscale videos from which optical flow can be extracted.

In designing the encoding patterns, correctly pairing up the spatial and spectral patterns and choosing the pattern sequence is critical for realizing a high compression ratio. In order to maximize the amount of information encoded in the limited number of measurements, we propose a mechanism where the spatial pattern and the spectral pattern on the DMD change simultaneously. Such joint spatial-spectral compression is different from previous research[11,12] where the two-stage spatial-spectral modulation model was used, keeping a spatial pattern unchanged while changing the spectral patterns. With a different spatial and a different spectral pattern for each modulation, different spatial components and different spectral components of the hyperspectral data are encoded. In this way, when recovering a hyperspectral video frame from a consecutive set of measurements, the amount of independent information carried in the measurements are maximized.

### Optical flow assisted 4DTV regularization for hyperspectral video reconstruction

To effectively and accurately recover the temporal-spatial-spectral data from compressive measurements, the additional redundancy or the higher sparsity in this 4D space needs to be fully utilized. An optical flow-assisted 4D Total Variation (4DTV) regularization model for hyperspectral video reconstruction is proposed as described by Eq. (3). It exploits the sparsity in the first or second-order derivatives of the data in all 4 dimensions to regularize the ill-posed inverse problem of compressive reconstruction.

$$ u = \arg\min(|\nabla_4 u| + |\partial_{t,OF} u|), \; s.t. |b - RSu| < \epsilon \qquad (3) $$

$$ where\, |\nabla_4 u| = \sum_{x,y,\lambda,t} \left( \sqrt{\left(\frac{\partial u_{x,y,\lambda,t}}{\partial x}\right)^2 + \left(\frac{\partial u_{x,y,\lambda,t}}{\partial y}\right)^2} + \left|\frac{\partial^2 u_{x,y,\lambda,t}}{\partial \lambda^2}\right| + \left|\frac{\partial u_{x,y,\lambda,t}}{\partial t}\right| \right) $$

$$ and\, |\partial_{t,OF} u| = \sum_{\substack{x,y,t1,t2 \\ |t1-t2| \in \{1,2\}}} |u_{x,y,t1} - u_{x+v_x, y+v_y, t2}| $$

The term $|\nabla_4 u|$ in Eq. (3) represents the expression for the 4DTV regularization. The widely used spatial total variation (TV) regularization approximates images with a piecewise constant model and assumes that images have sparse edges. Under this assumption, if an object in a video scene is moving, a pixel value will only change when an edge of the moving object passes through it. Consequently, the piecewise constant approximation can be naturally extended to the temporal domain[53-55]. For the sparsity in the spectral dimension, we apply the piecewise linear approximation given the properties of most spectra in the visible and near-infrared band and use the L1 norm of the second-order derivatives in spectral dimension for spectral regularization. Alternative spectral sparsity models, such as TV and L1 models and learned sparse representation dictionaries, can also be used for other spectral bands.

The term $|\partial_{t,OF} u|$ in Eq. (3) represents the optical flow constraint. Optical flow describes the distribution of apparent movement velocities of brightness patterns in images[56]. It has been proved to be very useful for the reconstruction of compressive video imaging[16,57-60]. However, the calculation of optical flow requires the knowledge of video frames while, on the other hand, the optical flow is expected to be used in video reconstruction for improved results. To solve this dilemma, we divide the reconstruction of a hyperspectral video into two stages. First, a grayscale video at the same spatial resolution is reconstructed from the raw measurements containing the complementary patterns (See "*Methods*"). More detailed are described in Section 2 of the Supplementary Information. Then, optical flow can be extracted between the frames of the reconstructed grayscale video and used in Eq. (3) for the final hyperspectral video recovery. Figure 2a shows the flow chart of the whole process of compressive measurement and reconstruction.

For two nearby frames at $t_1$ and $t_2$ in a video, if the optical flow vector field between them $\langle v_x(x,y), v_y(x,y) \rangle$ is provided, the constraint on the pixel intensities of these two frames can be written as $u_{x,y,t_1} = u_{x+v_x, y+v_y, t_2}$. It suggests that the content at location $(x,y)$ in frame $t_1$ has moved to $(x+v_x, y+v_y)$ in frame $t_2$. The optical flow provides an exact mapping of the pixels between nearby frames. Here, the optical flows between any two frames with a temporal distance closer than or equal to 2 are included in the TV term. The temporal TV term in the 4DTV algorithm only regularize the changes of values

between 2 nearest frames. In contrast, the optical flow constraint connects the data points in one frame to the data in its 4 nearest frames, 2 before and 2 after. Without the optical flow constraint, the 4DTV regularization itself will result in decreased reconstruction performance as demonstrated in the next section.

In addition, the ordering of the spatial STOne pattern sequence is designed in a 'structured random' way such that any consecutive $4k^2$ patterns in the sequence can be treated as a complete set of embedded $2k \times 2k$ STOne patterns, where k is a positive integer. With these properties of the STOne pattern sequence, multi-resolution reconstructions can be achieved from the same set of measurements. The embedded low-resolution patterns can be used to recover a low-resolution version of the hyperspectral video, with the corresponding 4DTV algorithm illustrated in Fig. 2b.

## Results of optical flow-assisted 4DTV reconstruction

The SDI system is calibrated and tested in experiments. The target scene as shown in Fig. 3a composed of a spinning color filter wheel with printed texts against a resolution chart. The color filter wheel is in focus of the SDI. Using the proposed joint spatial-spectral modulation scheme and the optical flow-assisted 4DTV reconstruction, 157 frames of hyperspectral video of size $128 \times 128$ with 64 spectral bands can be reconstructed from 184,000 single-pixel measurements, achieving a compression ratio of around 900:1. We did not push the pattern rate of the DMD and operate the DMD at 5 kHz. From the 5000 measurements during one second, 4.3 video frames are acquired. Details of system setup, calibration and modulation patterns can be found in Section 1 of Supplementary Information.

Remember, from the train of measurements, the data window selects and represents a block of measurements for recovering each video frame. Here, because the STOne patterns with multi-resolution property are used for spatial modulation, we use a data window of fixed length of 2048 and a sliding stride of 1024, meaning each frame is recovered from 2048 measurements and consecutive frames share an overlap of 1024 measurements.

During reconstruction, 157 frames of $128 \times 128$ grayscale video are first reconstructed by the 3DTV regularized algorithm, as demonstrated in Fig. 3b. On a laptop with an Intel Core i7-9750H CPU, the Matlab algorithm takes on average about 0.65 seconds to recover 1

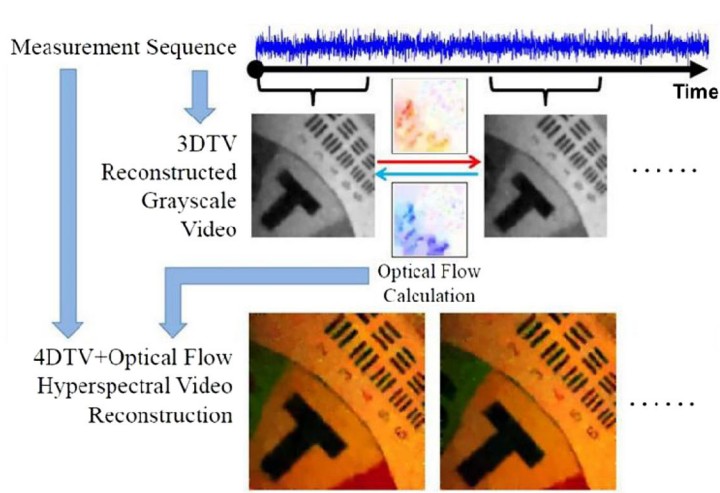

**Fig. 2 | Diagram and flow chart of the reconstruction process. a** Diagram illustrating the reconstruction process of the hyperspectral video from the measurement sequence of the SDI system, **b** Flow chart of the low-resolution hyperspectral video reconstruction algorithm using the embedded low-resolution STOne patterns.

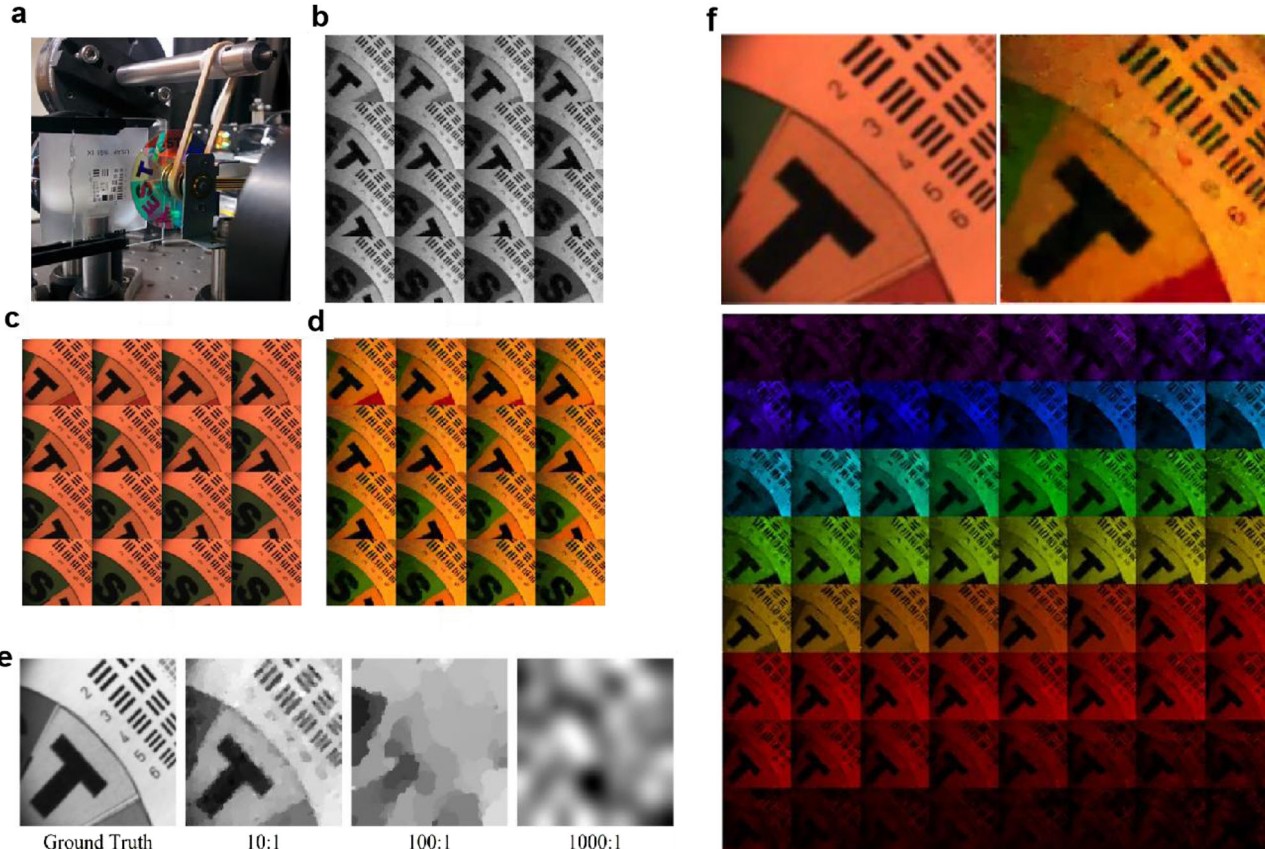

**Fig. 3 | Experimental reconstruction results of the SDI. a** The target scene imaged by the SDI, **b** 16 frames evenly selected from the 157 reconstructed grayscale video frames, **c** Frames recorded by a conventional color camera, **d** 16 frames evenly selected from the reconstructed 128 × 128 × 64 hyperspectral video frames with a compression ratio of 900:1 converted to artificial RGB images, e Reconstructions at CR of 10, 100, and 1000 if compression and sparsity are only used in the spatial dimension for a 128 × 128 grayscale image, **f** Top left: ground truth color camera capture. Top right: the corresponding reconstructed hyperspectral frame converted to artificial RGB image. Bottom: Expanded hyperspectral data cube of the frame on the top right, containing 64 images of 128 × 128 at different wavelength bands. The spectral resolution is 7.3 nm/band for the whole spectral range from 361 nm to 827 nm.

grayscale frame. Then the forward and backward optical flows are extracted from the grayscale video[59]. More details on the grayscale video and optical flow calculation are presented in Section 2 of Supplementary Information. With the joint spatial-spectral sensing patterns, the measurement values, and the optical flows, hyperspectral video is reconstructed by solving Eq. (3) using the Primal-Dual hybrid gradient (PDHG) solver[29,60]. The partial derivative operation of optical flow with respect to $t$ in the L1 term $|\partial_{t,OF} u|$ can be represented by a linear operator taking the difference between the corresponding pixels calculated from optical flow.

The use of sparsity in the 4D temporal-spatial-spectral space and the optical flow constraints are critical in realizing the 900:1 compression ratio (CR) and successfully reconstructing 4.3 frames of 128 × 128 × 64 hyperspectral data from only 5000 raw single-pixel measurements taken in one second. Simulated results in Fig.3e demonstrate the reconstructions if compression and sparsity are only used in the spatial dimension for the 128 × 128 grayscale image. For CR of 100:1 and above, a grayscale image cannot be reconstructed due to too few measurements. Because the data in the 2D space are far less sparse than in the 4D space, higher compression ratios lead to very poor reconstruction accuracy due to too few measurements.

At such a high compression ratio of 900:1, the Matlab algorithm of the PDHG solver takes about 390 seconds to reconstruct one hyperspectral frame on a laptop with an Intel Core i7-9750H CPU. When operating the DMD at 5 kHz, the data acquisition time results in 4.3 fps and reconstruction needs to be performed off-line. The long

reconstruction time of optimization-based algorithms motivates us to develop a deep-learning reconstruction approach introduced in the next section. We will show that deep learning greatly accelerates reconstruction and is much faster than the data acquisition rate of 4.3 fps, therefore enabling real-time imaging.

Figure 3d displays 16 hyperspectral frames from the reconstructed hyperspectral video. To simultaneously show the results of all spectral bands, we convert the hyperspectral images to sRGB via the CIE color-matching function. For comparison, images captured by a color camera, shown in Fig. 3c, illustrate the spatial accuracy of the reconstructions. Figure 3f displays the 64 single-band images of one reconstructed hyperspectral frame which are colored according to their wavelengths, along with a ground truth color camera image. The spectral resolution is 7.3 nm/band for the whole spectral range from 361 nm to 827 nm. In the hyperspectral data cubes, different color filters on the color wheel show different intensity changes over the 64 wavelength bands. The full 157 frames of the reconstructed grayscale and hyperspectral videos can be found in Fig. S6 in Supplementary Information and in Supplementary Video 1-3. Multiresolution reconstruction results based on the STOne patterns are presented in Section 2 in Supplementary Information and Supplementary Video 4-6.

Figure 4 plots the spectra of the pixels on different filters of the color wheel extracted from the reconstructed hyperspectral video and compares them with the ground truth spectra. The ground truth spectrum for each filter was directly measured by a fiber-coupled

spectrometer. Additional details of the full-resolution reconstruction and more analysis are presented in Section 3 of Supplementary Information.

Figure 5 compares the reconstructions from the same sets of measurements with and without using the optical flow constraints. The artificial RGB images recovered without optical flow in Fig. 5b lose color contrast compared to the ones in Fig. 5a. More apparent differences can be seen in the comparison of pixel spectra shown in Fig. 5c. For the pixels on the red and green filters, the spectra reconstructed without optical flow significantly deviate from the ground truth.

The spatial STOne patterns enable multi-resolution reconstruction from the same set of raw measurements. For example, 1024 measurements with $128 \times 128$ STOne patterns are enough to compose a full STOne transform embedded at $32 \times 32$ resolution. From the same set of measurements as the full-resolution reconstruction, we demonstrate the results of a low-resolution reconstruction of $32 \times 32$ spatial pixels with 64 spectral bands at 4.3 fps, using a data window of length 1024 and a sliding stride of 1024. Grayscale videos at this resolution can be calculated by a simple linear inverse transform without any iterative operations. The reconstructed $32 \times 32$ grayscale videos are illustrated in Fig. 5d and the calculation takes 0.5 ms per frame in Matlab. This method is useful in getting a quick look at the spatial information captured by the SDI. The $32 \times 32 \times 64$ low-resolution hyperspectral video reconstruction as shown in Fig. 5e takes 45 seconds per frame.

**Deep neural networks for hyperspectral video reconstruction**

We also develop a deep learning reconstruction approach to address the challenge of the long reconstruction time of optimization-based algorithms. The 4D temporal-spatial-spectral correlation in the signal is embodied and incorporated in the neural network design. In the 4DTV algorithm, the grayscale video is first reconstructed to provide a temporal correlation. Similarly, we adopt such Divide-and-Conquer method in the deep learning approach to first reconstruct grayscale videos which are used as additional information in hyperspectral frame reconstruction. As demonstrated in Fig. 6, the deep learning approach is composed of two stages: first, a model based on the long short-term memory (LSTM) network reconstructs the grayscale video from CS measurements leveraging temporal correlation among 5 adjacent frames; secondly, a hyperspectral reconstruction network based on convolutional neural networks (CNNs) with residual connections recovers hyperspectral frame from the joint spatial-spectral compressed measurements together with the grayscale video reconstructed in the first stage.

The grayscale video reconstruction network is comprised of two modules. The first module is a CNN with residual connections for initial video reconstruction from the spatial CS measurements, which can be acquired from pairs of spectral complementary measurements (See "*Methods*"). As such, the measurement equation for the grayscale video can be derived from Eq. (1) as $y_t = \Psi_s x_t$, where $x_t$ is the vectorized grayscale video at time $t$, $\Psi_s$ represents the sensing matrix for the pure

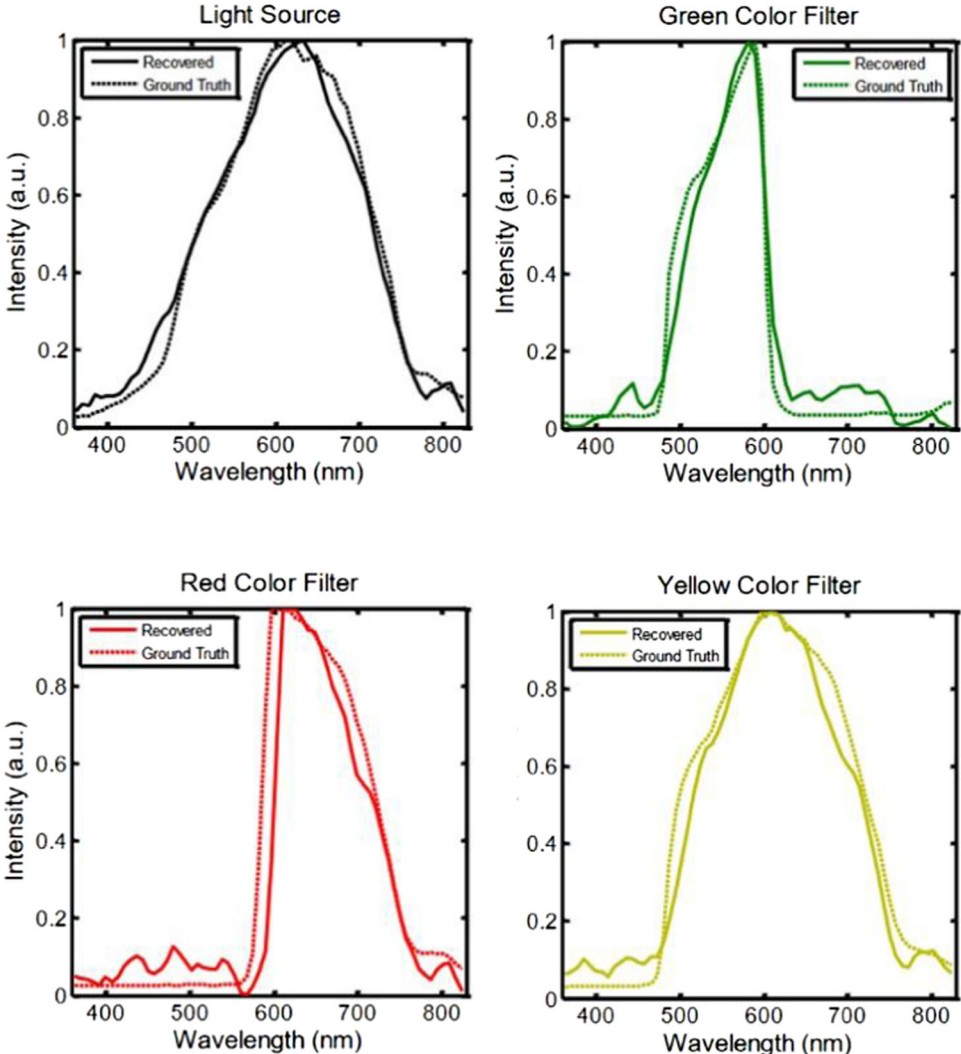

**Fig. 4 |** Reconstructed and ground truth spectra of the pixels in the hyperspectral video on the light source and on the green, red, and yellow color filters of the color wheel.

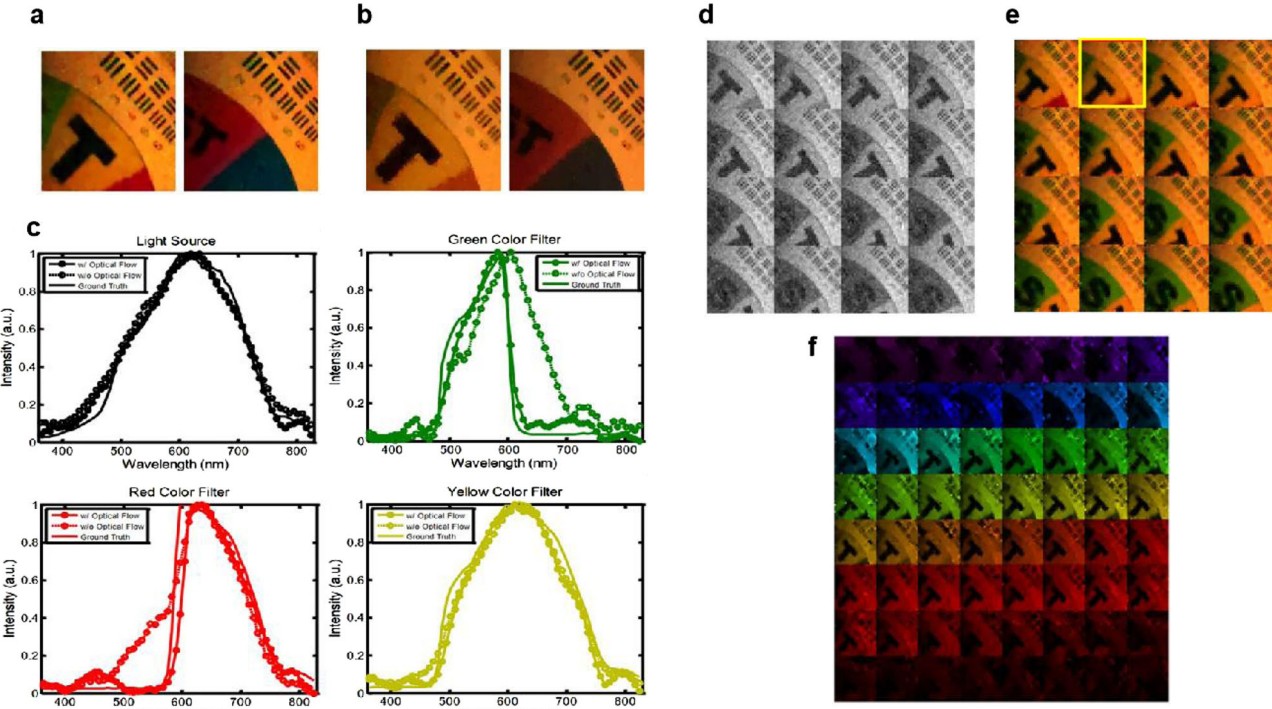

**Fig. 5 | Reconstruction with and without optical flow regularization and low-resolution reconstruction results. a** Artificial RGB images of reconstructions with optical flow. **b** Artificial RGB images of reconstructions without optical flow. **c** Spectra of different pixels from the hyperspectral videos reconstructed with and without optical flow. **d** 16 frames evenly selected from the 157 frames of the 32 × 32 L2-reconstructed grayscale video, **e** 16 frames evenly selected from the 157 frames of the 32 × 32 × 64 low-resolution hyperspectral video converted to RGB images, **f** Expanded hyperspectral data cube of the frame marked in yellow square in **e** containing 64 32 × 32 images at different wavelength bands. The spectral resolution is 7.3 nm/band for the whole spectral range from 361 nm to 827 nm.

spatial measurements, which can be acquired by removing the rows not used in sensing in $(\Phi_s)^T$. The vector $(\Psi_s)^T y_t$ after being reshaped into 2D is used as input to the network which outputs the reconstructed grayscale video. The CNN module is pre-trained to improve reconstruction quality. The CNN module is based on several identical blocks named the "RC block" (See "*Methods*"). The second module is an LSTM network taking in 5 adjacent grayscale frames each reconstructed from the same CNN module and outputs 5 video frames enhanced by temporal correlation and implicit motion estimation between the frames learned by the LSTM network (see "*Methods*").

The hyperspectral frame reconstruction network has the same backbone as the CNN module composing of identical RC blocks but with different input and output sizes. In CS measurement process, Eq. (1) can be reformulated as $\tilde{b}_t = \tilde{\Phi} u_t$. Here, $\tilde{\Phi}$ represents the matrix where the rows of zeros in matrix $R_t \tilde{\Phi}$ are removed, $\tilde{b}_t$ is the measurement vector where the empty entries in $b_t$ are removed. We calculate the vector $(\tilde{\Phi})^T \tilde{b}_t$, reshape it into the size of the 3D hyperspectral frame to be reconstructed, concatenate it with the corresponding frame from the reconstructed grayscale video along the spectral dimension, then use such 3D tensor as input to the hyperspectral frame reconstruction network which outputs the reconstructed frame.

## Results of deep neural networks for hyperspectral video reconstruction

We present the reconstruction results of the proposed deep learning approach on simulation data and on experimental data. Reconstruction time comparison with the optical flow-assisted 4DTV optimization approach is also presented.

First, we present the results of testing on simulation data. We adopt the strategy of creating hyperspectral video datasets from hyperspectral image by circularly shifting the images as elaborated in

Section 4A of Supplementary Information. The hyperspectral image datasets used are CAVE dataset[61] and Harvard dataset[62]. Hyperspectral video blocks of size 32 × 32 with 31 spectral bands and 5 temporal frames are extracted from such videos for training and testing due to GPU memory limit. Simulated compressive measurements are taken on the video blocks. We train the base network models on a single NVIDIA Geforce RTX 3070 GPU with 8GB memory. Table 1 shows the mean of the reconstruction quality measured in the peak signal-to-noise ratio (PSNR), structural similarity (SSIM)[63], and spectral angle mapping (SAM)[64] for the testing dataset without adding measurement noise for CR of 100, 25, and 10. The metric is calculated on every spectral channel and averaged over all spectral channels of all frames. Larger values of PSNR and SSIM suggest better performance, while a smaller value of SAM implies a better reconstruction. Figure 7 illustrates an example reconstructed hyperspectral frames using the deep learning approach for CAVE and Harvard datasets. Each image is composed of 6 × 6 non-overlapping tiles of 32 × 32 × 31 reconstructed hyperspectral blocks converted to RGB image using CIE color mapping function. More results are shown in Figure S7 and S8 of Supplementary Information and in Supplementary Video 7 and 8.

Figure 7c, d shows the absolute error between ground truth and reconstruction results along spectral dimension. Each ground truth 32 × 32 × 31 hyperspectral block has been normalized to have pixel values between 0 and 1. The absolute error for each spectral band is averaged across spatial dimension, temporal dimension, and all test data. With a given resolution, a lower CR means a greater number of measurements which embed more information of the scene than higher CR, and therefore produces higher spectral accuracy.

Since noise is inevitable in real measurements, we conduct additional groups of experiment to analyze the effect of adding measurement noise to simulation data. Noise analysis is performed by fine-tuning the base models to simulate real scenarios. More details of

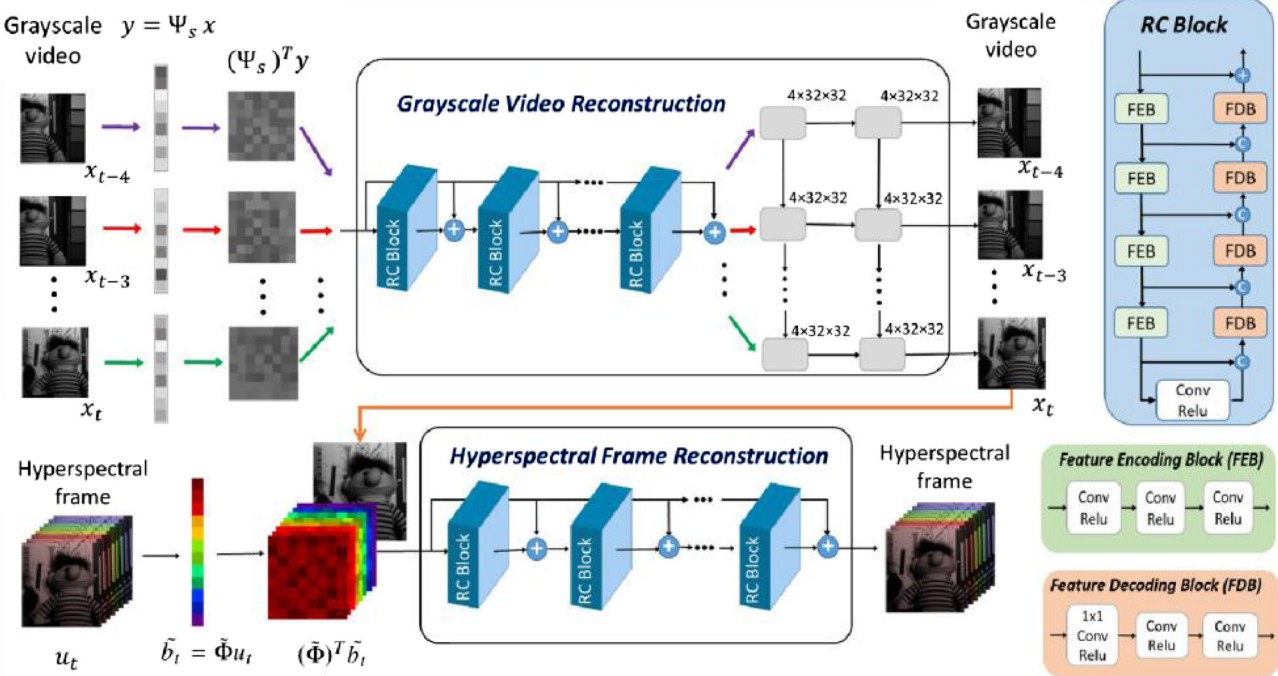

**Fig. 6 | Schematic of the deep learning approach for hyperspectral video reconstruction.** First, the grayscale video of 5 frames is reconstructed from spectral complementary measurements through a CNN module composed of RC blocks and an LSTM network module. Then, the joint spatial-spectral compressed measurements after pre-processing are concatenated with the reconstructed grayscale video frame, which serves as the input to the hyperspectral frame reconstruction network. The "C" with a circular block denotes the concatenation, the "+" with a circular block denotes the summation. The Feature Encoding Block (FEB) and Feature Decoding Block (FDB) are composed of 3 convolutional layers each followed by a ReLU. The "Conv" refers to a convolutional layer of kernel size 3 × 3 except in the first layer of the FDB where the kernel size is 1 × 1.

model training and results of noise analysis are presented in Section 4A of Supplementary Information.

We also test the deep learning approach on experimental data from the SDI to further validate its performance. Since the SDI has 64 spectral bands at 7.3 nm/band from 361 nm to 827 nm and no publicly available hyperspectral dataset has the same spectral bands, we adopt the strategy of interpolating the Harvard dataset in the spectral dimension to have the same spectral bands as the SDI between 423 nm and 715 nm, leading to 41 bands. The neural network models are trained based on the interpolated dataset with Gaussian measurement noise added. Due to GPU memory limit, the models are designed to recover hyperspectral image patches of spatial size 32 × 32 with 41 spectral bands at 7.3 nm/band from 423 nm to 715 nm (see "*Discussion*"). Model architecture and training scheme are the same as described in Section 2.5. To match the size of the model input, we divide the full spatial view of the SDI of 128 × 128 into 16 blocks of 32 × 32. Compressive measurements are taken consecutively for each block within every video frame. The blocks are individually reconstructed by the network models and stitched together to form a full hyperspectral image of 128 × 128 with 41 spectral bands. Details of data generation, model training, and model testing are described in Section 4B of Supplementary Information.

The same target scene as shown in Fig. 3a is used for testing. Figure 8a displays 4 frames from the reconstructed hyperspectral

video for each CR converted to artificial RGB images along with corresponding color camera capture. The full reconstructed hyperspectral videos can be found in Fig. S9 and Supplementary Video 9-11. Figure 8b plots the spectra of the pixels on different filters of the color wheel extracted from the reconstructed hyperspectral video and compares them with the ground truth spectra. The reconstruction of lower CR shows better spectral accuracy in general than higher CR since more information about the scene is collected with more measurements. To enhance the reconstruction performance, more scenes similar to the color filters can be included in the training dataset since Harvard is an outdoor scene dataset. A more realistic noise model will also help with the performance.

Table 2 demonstrates the reconstruction time comparison in seconds between the proposed optical flow-assisted 4DTV regularization approach for a 128 × 128 × 64 hyperspectral frame and the deep learning approach for 16 of 32 × 32 × 41 hyperspectral patches at CR100/CR25/CR10 on CPU (Intel Core i7-9750H) and on GPU (NVIDIA GeForce RTX 3070 with 8GB memory). The 4DTV-based optimization algorithm is not tested on GPU due to its iterative nature. As can be seen, the deep learning approach enjoys significant gains in reconstruction time. It can reconstruct at least 16 frames per second, allowing for real-time reconstruction when data acquisition is slower. Remember, the data acquisition time for the experiment of color filter wheel is 4.3 fps. For comparison, using the optimization algorithm, real-time imaging is not possible and reconstruction needs to be performed off-line. In Table 2, we display the sum of reconstruction times for 16 patches in order to compare with the optimization algorithm at the same spatial resolution. However, in reality, the reconstruction of 16 patches can be parallelized because there is no dependency between the patches, allowing for reconstruction at 16 times higher frame rate. The CR can be chosen based on factors including the expected reconstruction quality, data acquisition time, reconstruction time, etc.

**Table 1 | Quantitative evaluation of PSNR(dB)/SSIM/SAM over test data without measurement noise added**

| CR | CAVE | Harvard |
|---|---|---|
| 100 | 23.15/0.703/0.159 | 24.86/0.712/0.083 |
| 25 | 24.06/0.736/0.138 | 25.69/0.747/0.074 |
| 10 | 28.03/0.838/0.086 | 28.25/0.841/0.063 |

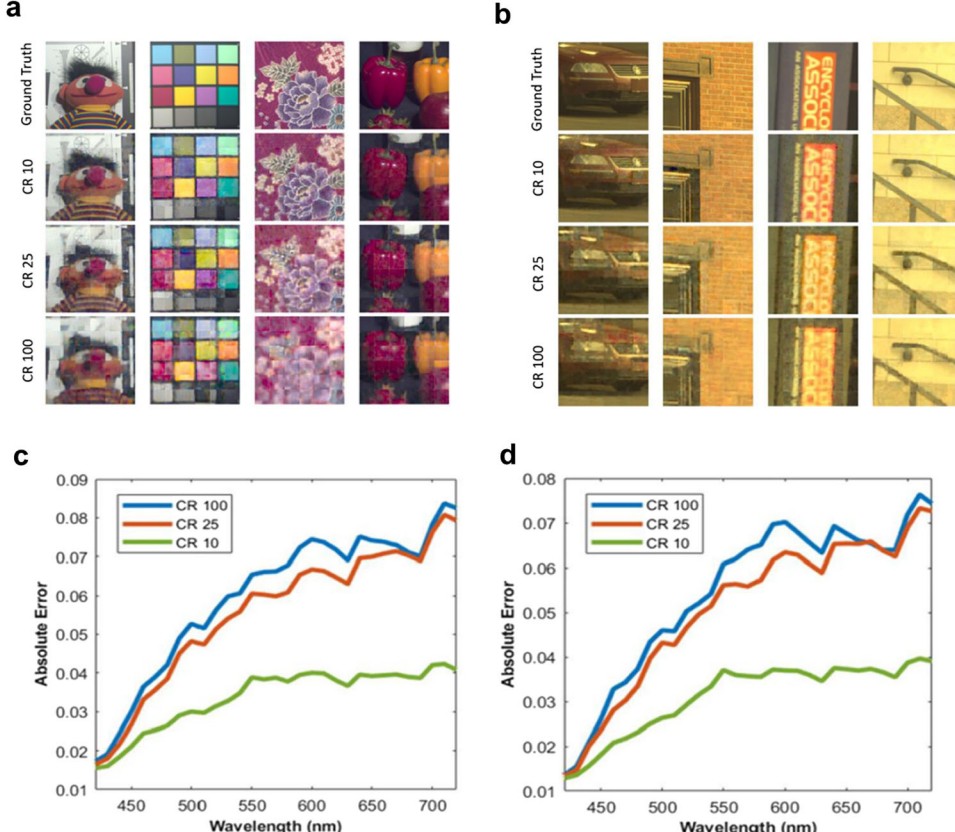

**Fig. 7 | Reconstruction results of the deep learning approach on publicly available datasets.** Example reconstruction results of the deep learning approach of **a** CAVE dataset and **b** Harvard dataset. A gamma correction of gamma = 0.4 is applied on the converted RGB image to brighten darker areas. The first row is ground truth, the rest three rows are for CR of 10, 25 and 100, **c** The absolute error between ground truth and reconstruction results of the proposed deep learning approach along the spectral dimension for CAVE dataset, and for **d** Harvard dataset.

## Discussion

In summary, combining a compressive encoding scheme and a reconstruction algorithm that exploits the inherent redundancy in the 4D hyperspectral data with a unique optical system based on a single-pixel detector, we report the SDI system that can achieve high-throughput hyperspectral video recording at a low bandwidth. The system is demonstrated to capture hyperspectral video of size 128 × 128 with 64 spectral bands at 4.3 fps offering a 900× data throughput compared to conventional imaging, which is a first-of-its kind of a single-pixel-based hyperspectral imager. Other features such as multi-resolution reconstruction are also realized and demonstrated with the experimental data. To address the challenge of the long reconstruction time of the optimization-based approach, a deep learning approach for fast recovery has also been developed for CS hyperspectral video reconstruction of the SDI on simulation data and experimental data.

We discuss the advantages and novelty of the proposed single-pixel approach for hyperspectral video imaging over existing methods reported in the literature First, by eliminating the need for a 2D array sensor (e.g., CMOS or CCD), single-pixel approach can be easily extended for imaging outside the visible wavelengths such as infrared, terahertz (THz), and X-ray imaging[6–9,65], for which the 2D sensor is either of low resolution, prohibitively expensive, or lacks cutting-edge performance of a single pixel sensor. Secondly, single-pixel imaging may use exotic detectors whose 2D format with high specialization are impractical to manufacture. For instance, employing an ultrasonic transducer as detector, SPI has been used for photoacoustic imaging[66,67]. SPI paired with the photomultiplier tube has found applications in single-photon imaging[68], for time-of-flight 3D imaging[69,70], and imaging objects hidden from direct linefa-of-sight[71]. Thirdly, directly recording hyperspectral video of such high dimensionality requires large storage, significant transmission bandwidth as well as high power consumption, memory footprint, and time cost, imposing extreme pressure on imaging systems especially when storage and transmission of the data is critical, e.g., on satellites, and rovers where the sampled data typically needs to be transmitted to another facility for processing or reconstruction. The high compression ratio achieved in the SDI allows transmission of high throughput data at a low bandwidth significantly relieving the pressure on bandwidth and storage. It is also useful, for instance, in biological microscopy for long term observations of certain bioprocesses which occur sporadically by greatly saving on data storage. As discussed in the Introduction section, the 900:1 sampling compression ratio is much higher than both the 2D detector-based and the single pixel-based techniques in current literature at similar reconstruction quality, endowing the SDI system with unique benefits to be used in resource-constraint imaging systems.

The fundamental contribution or innovation that enables the high compression ratio or data throughput in the SDI system is the proposed framework which fully exploits signal sparsity in 4D space in both sensing and recovery: the encoding scheme proposed allows the spatial-spectral information to be jointly and maximally acquired through a small number of measurements and the temporal correlation (optical flow) can be extracted from the same raw measurements, consequently embedding more scene information in each measurement; the signal prior of joint sparsity in 4D space are proposed and realized in the reconstruction, extracting more information from the compressed measurements.

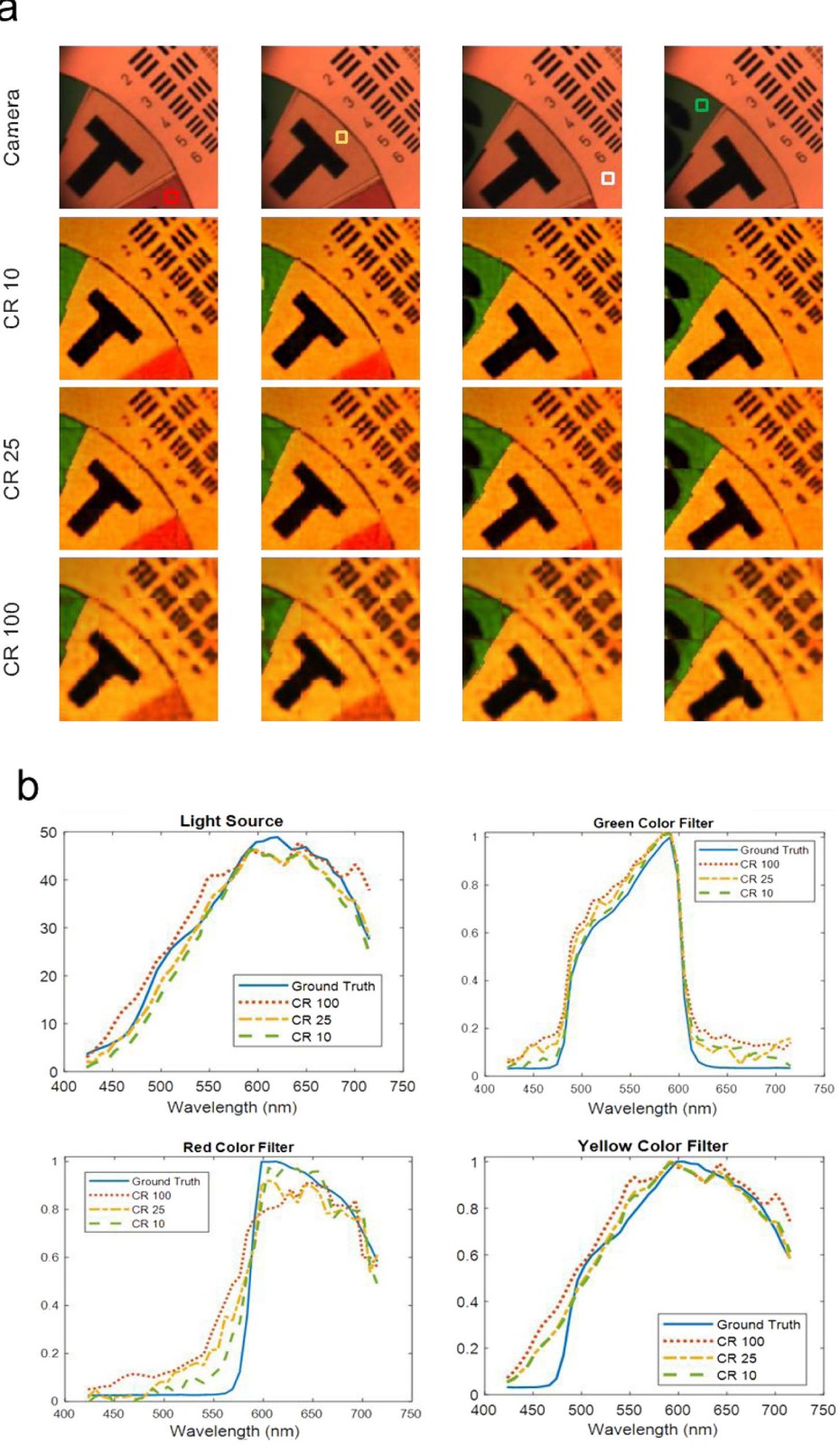

**Fig. 8 | Reconstruction results using the deep learning approach testing on experimental data of the SDI. a** Top row: Frames recorded by a conventional color camera. Other rows: 4 example frames from the reconstructed 128 × 128 × 41 hyperspectral video frames converted to artificial RGB images at CR of 10, 25, and 100. **b** Reconstructed and ground truth spectra of the example pixels in the hyperspectral video. The example pixels are marked by color squares in the top row in **a** for the red filter, the yellow filter, the light source, and the green filter, respectively, from left to right in the top row.

**Table 2 | Reconstruction time of the 4DTV-based optimization approach for reconstructing one 128 × 128 × 64 hyperspectral frame and the deep learning approach for reconstructing sixteen of 32 × 32 × 41 hyperspectral patches at CR 100/CR 25/CR 10 on the same CPU and GPU**

| Reconstruction approach | CPU time (s) | GPU time (s) |
|---|---|---|
| 4DTV-based Optimization | 390 | – |
| Deep Learning (16 patches) | 1.9296/1.6992/1.2112 | 0.0624/0.048/0.0368 |

For the SDI system, it is straightforward to simultaneously increase all of the spatial resolution, spectral resolution, and frame rate. Besides, with a resolution increase in one or more dimensions, the signal becomes sparser and an even higher compression ratio can be achieved at the same reconstruction quality. With the DMD as the light modulator, the spatial resolution can be much larger than 128 × 128. The current SDI prototype can have at least 256 spectral channels and we choose 64 since it is enough for the experiment. The spatial and spectral resolutions are constraint by the DMD resolution, the expected measurement noise, computation capacity, etc. There will be a limit where signal strength compared to noise will make each measurement indistinguishable in the detector output and the A/D encoding and thus result in the inability to accurately reconstruct the signal. Although increasing the spatial and/or spectral resolution may lead to decreased SNR of the measurements, with improved reconstruction algorithms, e.g., neural networks trained with the corresponding noise level, or using another denoising network, the same reconstruction quality could be obtained at a higher resolution and higher compression ratio. We did not intent to push the limit of the spatial or spectral resolution in this paper but aim to propose an architecture and algorithms which fully exploits the signal sparsity in 4D space in a single-pixel setting.

The frame rate demonstrated is also far from reaching its limit. As described before, we can "slide" the "data window" along the train of single-pixel measurements to select a consecutive block of measurements to define and recover each video frame. Using a sliding stride of 1 will result in the maximum frame rate. In the color wheel experiment, the data window length is 2048 measurements and the sliding stride is 1024. We did not push the limit of the frame rate by using a smaller sliding stride because then the content of nearby frames would have minor difference since the change in motion is far slower than the period of one measurement. We chose a sliding stride suitable for the motion in the scene. Even if increasing the spatial or spectral resolution may require a data window with larger length, the sliding stride can always be independently chosen and thus the maximum frame rate is, in theory, not affected by the change in spatial or spectral resolution. However, for imaging faster motion, the number of spatial-spectral patterns being grouped per frame would have to be reduced, as discussed in our earlier CS-MUVI publication[72] and correspondingly may reduce the spatial/spectral pixel count to maintain the same quality.

In addition, we did not operate the DMD at its maximum pattern rate. As discussed in Introduction, one type of SPI techniques aims to achieve ultra-high frame rate by mechanically boosting the modulation pattern rate but at the cost of low reconstruction quality due to low measurement SNR[48,50]. These methods did not exploit the inherent redundancy of the signal which allows for reducing sampling without losing information or measurement SNR. On the other hand, however, by combining such fast alternatives to the DMD with the SDI system, ultra-high frame rate hyperspectral imaging may be realized. However, high pattern modulation rates lead to noisy measurements which will get more severe when it comes to spectral imaging. Therefore, reconstruction quality would be one of the key problems to tackle for such systems if realized. In the deep learning reconstruction for experimental data, the neural network models are designed to recover

hyperspectral image patches of spatial size 32 × 32 with 41 spectral bands due to GPU memory limit. The temporal module has around 6 million parameters and the hyperspectral image reconstruction network has around 1 million parameters. The memory bottleneck, however, resides in the multiple RC blocks in the networks where each RC block produces and stores many intermediate feature maps. More optimized network designs or algorithm implementation could help reduce memory cost. We believe that with hardware of higher capacity, the deep learning approach can be used to recover hyperspectral video frames of larger sizes. The test data is acquired by the SDI with a block-based strategy where the full spatial view of 128 × 128 is divided into 16 blocks of 32 × 32. Compressive measurements are taken consecutively for each block within every video frame. We demonstrated a high CR of 100:1 when reconstructing the 32 × 32 × 41 hyperspectral image patches. Because the block-based measurement strategy must be adopted in this case to match the smaller input size of the neural network models, we do not aim nor expect the same high CR of 900:1 to be achieved as when reconstructing the 128 × 128 × 64 hyperspectral frames demonstrated in the 4DTV-based reconstruction for two reasons. First, signal sparsity generally decreases with smaller spatial, spectral and temporal size and therefore the signal is less compressible than a signal of larger size, leading to a lower compression ratio if to obtain the same reconstruction quality as a signal of a larger size. Secondly, because the 16 blocks are measured consecutively within every video frame, the strategy of using an overlapping data window when measuring one block is not applicable. The block-based measurement strategy is only a temporary effort to match the input size of network models limited by the GPU memory. We believe hyperspectral frames of larger sizes at much higher compression ratios can be reconstructed by the deep learning approach with hardware of higher capacity. We would like to leave it as our future work in this direction. As the initial attempt to incorporate the 4D temporal-spatial-spectral correlation of the signal in the neural network design for CS hyperspectral video reconstruction in the single-pixel setting, the results demonstrate the feasibility of a fast reconstruction solution and promising reconstruction quality.

## Methods
### Complementary patterns
In Eq. (1), The binary modulation matrices $\Phi_s$ and $\Phi_w$ have positive and negative values, while light modulation with the DMD can only realize 0-1 modulation. In order to match the mathematical model, spatial and spectral complementary patterns are inserted into the pattern sequence. For a joint spatial-spectral measurement, we have

$$y_t = \varphi_s^T * X_t * \varphi_w = \left(\varphi_s^+ - \varphi_s^-\right)^T * X_t * \left(\varphi_w^+ - \varphi_w^-\right) \tag{4}$$

Here, $X_t$ is the $N \times K$ matrix representing the hyperspectral frame at time $t$, $\varphi_s$ is an $N \times 1$ vector representing one of the spatial modulation patterns during measurement of $X_t$, $\varphi_w$ is an $K \times 1$ vector representing the joint spectral pattern of $\varphi_s$, $y_t$ is the single pixel measurement. Both $\varphi_s$ and $\varphi_w$ are binary patterns with $\{+1, -1\}$ entries. Because the DMD cannot directly display negative entries, we split $\varphi_s$ into positive part and negative part as $\varphi_s = \varphi_s^+ - \varphi_s^-$, where $\varphi_s^+ = \max(\varphi_s, 0), \varphi_s^- = -\min(\varphi_s, 0)$. As such, we have $\varphi_s^+ + \varphi_s^- = I_s$, where $I_s$ is the $N \times 1$ vector with all 1 entries. We call $\varphi_s^-$ the spatial complementary pattern for $\varphi_s^+$. Likewise, $\varphi_w$ can be split as $\varphi_w = \varphi_w^+ - \varphi_w^-$, and $\varphi_w^-$ is called the spectral complementary pattern for $\varphi_w^+$. We display $\varphi_s^+$ and $\varphi_w^+$ on the DMD in place of $\varphi_s$ and $\varphi_w$, respectively. In the experiment of color filter wheel target in Section 2.4, one spectral complementary pattern $\varphi_w^-$ is inserted for every spectral modulation $\varphi_w^+$, which means for every measurement of $\left(\varphi_s^+\right)^T * X_t * \varphi_w^+$, we keep the spatial pattern unchanged and display $\varphi_w^-$ to obtain the measurement of $\left(\varphi_s^+\right)^T * X_t * \varphi_w^-$. At the same time, one spatial complementary pattern $\varphi_s^-$ is inserted for every 512 different

spatial modulations. It means for every 512 pairs of original and spectral-complementary measurements, we take two extra measurements where we use the spatial complementary pattern $\varphi_s^-$ corresponding to the first spatial pattern $\varphi_s^+$ to measure $(\varphi_s^-)^T * X_t * \varphi_w^+$ and $(\varphi_s^-)^T * X_t * \varphi_w^-$. Since $\varphi_s^+$ and $\varphi_s^-$ are both random patterns with half of entries being 1 and the other half being 0, the term $\varphi_s^- * X_t$ produces approximately half of the sum of all pixel intensities for each spectral channel. Assuming the target scene does not change too fast such that its overall intensity stays almost the same during every 512 different spatial modulations, we take the strategy of using the same $\varphi_s^-$ to serve as the complementary pattern for all the 512 different spatial patterns, which produces a well-approximated result as using the true $\varphi_s^-$ for each different spatial pattern. In this way, the four terms in Eq. (4), i.e. $(\varphi_s^+)^T * X_t * \varphi_w^+ + (\varphi_s^+)^T * X_t * \varphi_w^- + (\varphi_s^-)^T * X_t * \varphi_w^+ + (\varphi_s^-)^T * X_t * \varphi_w^-$ can all be obtained which produces the measurement value $y_t$ for the joint spatial-spectral measurement $\varphi_s^T * X_t * \varphi_w$. In addition, because $\varphi_w^+ + \varphi_w^- = I_w$ where $I_w$ is an all 1 vector, the sum of every pair of original and spectral complementary patterns $\varphi_s^T * X_t * (\varphi_w^+ + \varphi_w^-)$ produces a pure spatial modulation result for calculation of optical flow of the grayscale video.

### RC block and the LSTM network

The so-called "RC block" in the reconstruction neural networks contains the feature extraction part and image reconstruction part connected by a convolution layer. The feature maps generated during feature extraction are concatenated with those generated in the image reconstruction part. Here, concatenation is used here instead of summation according to experimental results. The feature extraction part comprises of four feature encoding blocks (FEBs). Each FEB contains three convolutional layers of $3 \times 3$ kernel each followed by a rectified linear unit (ReLU). Each convolutional layer generates 64-channel feature maps. The image reconstruction part contains four feature decoding blocks (FDBs). Each FDB consists of three convolutional layers, where the first layer has $1 \times 1$ kernels and the other two layers has $3 \times 3$ kernels. Each convolutional layer generates 64-channel feature maps followed by ReLU. For the CNN module in grayscale video reconstruction network and for the hyperspectral frame reconstruction network, we use 11, 8, and 6 RC blocks for compression ratios of 100, 25, and 10, respectively.

We enforce the RC block to predict the residual by adding a residual connection that adds the block input to the block output, since residual learning enables fast and stable training. We do not use any pooling layer or up-sampling layer to avoid losing image details. Different RC blocks in the same network are enforced to share the same parameters to avoid over-fitting. Besides, we introduce the enhancement of additional residual connections which add the network input to the output of every RC block to utilize the input information in every stage of the network information flow for improved reconstruction quality. Furthermore, instead of using the simple residual connection, we propose the improvement of adding a weight to it. Therefore, every residual connection in the networks has an individually learnable weight, which increases network flexibility while imposing little computation burden. The long-short-term memory (LSTM) network module takes as input the output of the same pretrained CNN module for 5 adjacent videos frames $x_{t-4}, x_{t-3}, x_{t-2}, x_{t-1}, x_t$ and outputs 5 reconstructed grayscale video frames. The LSTM network has two hidden layers. The size of output of each hidden layer is $4 \times 32 \times 32$. The LSTM network learns the temporal correlation and implicit motion estimation between the 5 frames and outputs enhanced video frames based on the CNN module output.

### Data availability

The datasets used for the deep learning approach are publicly available from the following sources. The CAVE dataset can be obtained at https://www.cs.columbia.edu/CAVE/databases/multispectral/. The Harvard dataset can be obtained at http://vision.seas.harvard.edu/hyperspec/d2x5g3/.

### Code availability

The code for the reconstruction algorithms in this study is available from the corresponding author upon reasonable request.

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

## Acknowledgements

This research has been supported by the W. M. Keck Foundation and the National Science Foundation (NSF) (Grant #CHE-1610453). The authors thank Prof. Richard G. Baraniuk from Rice University for helping bring about collaboration between the authors.

## Author contributions

Y.X. conceived and developed the deep learning reconstruction framework including principles and algorithms, designed and performed relevant experiments and simulations, analyzed data, and wrote the manuscript. L.L. developed the 4DTV reconstruction approach, performed relevant experiments and simulations, analyzed data, and contributed to the first draft. K.K. conceived and initiated the research. V.S. contributed to discussion and the preparation of the manuscript.

## Competing interests

The authors declare no competing interests.
