## [Peer Review File · Nature Communications]

A Compressive Hyperspectral Video Imaging System Using a Single-Pixel DetectorREVIEWER COMMENTS

Reviewer #1 (Remarks to the Author)

This study proposes and demonstrates the use of single pixel imaging for video rate hyperspectral imaging.). To do this, the authors present an optical design that enables joint spatial-spectral encoding using a single digital micromirror device and a single point detector, where traditional single-pixel approaches rely on either spatial or spectral encoding or multiple detection arms. I found the paper very interesting and compelling. The proposed encoding scheme is clever, as spatial-spectral encoding allows better exploitation of the compressibility of temporal hypercubes, thus reducing the number of measurements and allowing faster acquisition. To the best of my knowledge, this approach is new and is likely to influence the field of single-pixel imaging and, more generally, computational imaging. Other notable results include the use of an optical flow assisted reconstruction algorithm that estimates the optical flow from the raw measurements. The authors convincingly demonstrate their claims and conclusions by imaging a rotating colour wheel at more than 4 fps ($128 \times 128 \times 64$ hypercube with a spectral resolution of ~ 7 nm).

Comments on the main document:

- I strongly recommend discussing references A, B, C, D and E in the introduction and/or discussion. Ref A proposes an adaptive method that can drastically reduce the number of measurements required for hyperspectral imaging. Ref. B proposed another adaptive method that allows hyperspectral video with a spatial resolution of 600×900 pixels, a spectral resolution of 10 nm over visible wavebands, and a frame rate of 18 fps. Both methods rely on similar but different reconstruction assumptions and hardware. Refs. C and D introduced fast alternatives to DMDs for single pixel modulation. Although not specifically designed for spectral imaging, they could be extended to spectral imaging. In particular, it would be interesting to discuss whether and how such modulation strategies could be combined with the proposed methodology. Reference F describes an ecosystem for "standard" single pixel hyperspectral imaging, where $128 \times 128 \times 2048$ hypercube are acquired in 12s. It also presents a database of more than 150 hypercubes.
- Line 75: "Single-pixel imaging is an emerging computational imaging technique". I would say that single-pixel imaging is now an established technique.
- Line 123: "Soldevila et al. reported a compression ratio of 3000:1". Compression ratios often carry little information in the sense that the image dimension (i.e. number of pixels) used for reconstruction can be chosen arbitrarily. For a given number of measurements, e.g. $M = 1,024$, the image can be reconstructed by choosing $N = 4,096$ pixels, resulting in a compression ratio of 4:1, or by choosing $N = 16,384$ pixels, resulting in a compression ratio of 16:1. Therefore, two reconstructions with the same spatial resolution (in terms of the ability to distinguish between two nearby objects) can have arbitrarily different compression ratios.
- Line 208: of "?? spatial patterns and ?? spectral"
- Line 230: "In practice, the data windows for each frame may overlap or have different widths." I think this sentence needs clarification. Does this "width" refer to the "sliding window" mentioned in line 349? As I am probably missing something here, I do not see the interest in introducing equation (2), which brings no additional information. Unless you just want to define b and R for equation 3?
- Line 250: "Both are binary patterns that can be easily implemented on the DMD". I would refer to the section in the supplementary document where you explain how to handle the negative value, as there are several variants.

- Equation (3). To reconstruct the dynamic hypercube (x,y,λ,t) , the authors consider a 4D TV prior (isotropic TV in space, anisotropic TV over the temporal and spectral dimensions). The spectral prior favours piecewise linear spectra, while the temporal prior favours piecewise constant dynamics. Could you motivate such a prior? Why not just enforce smooth variations across the temporal and spectral dimensions? This should be less computationally demanding. The same question applies to the L1-based optical flow prior, which allows for localised motion errors.
- Line 310: “However, the calculation of optical flow requires the knowledge of video frames while, on the other hand, the optical flow is expected to be used in video reconstruction for improved results”. I would recommend mentioning this point earlier, typically right after equation (3), which introduces v_x and v_y , and explaining that a method for estimating the vector field from the raw measurement will be introduced later. I believe that the current description of this "trick" (lines 313-329) is not clear enough, in contrast to section 4.1 and sections 1 and 2 of the supplementary document, which describe very well what is going on. I would suggest deleting lines 313-329 (and Figure 3) and referring the reader to Section 4.1 and the Supplementary Document. An alternative would be to improve the paragraph, where it may be more important to focus on the idea of complementary patterns giving access to the greyscale video, rather than the idea of "preview" (which could be removed for simplicity?).
- Line 347: “In practice the data windows for each frame may overlap, or be of different widths”. As mentioned above, I have not been able to understand what the “width” refers to and how
- Line 420: How does the reconstruction approach compares to the one described in Ref. F? What are the main differences?
- Line 430: “First, the grayscale preview video of 5 frames is reconstructed from spectrally complementary measurements”
- Line 523: “to GPU memory limit, the models are designed to recover hyperspectral image patches of spatial size 32×32 with 41 spectral”.
 - Where is the memory bottleneck of the proposed deep learning approach? How many learnable parameters do you have in the “temporal” module? In the “hyperspectral” module?
 - As shown in Fig. S9, processing small patches leads to discontinuities at the boundary of the patches. How do you correct for these artefacts? Have you considered the reconstruction of “moving” patches that you can merge using another (small) CNN?
- Line 628. “After the measurements, calibration values are calculated from these complementary patterns, and are subtracted from the measurement outputs to get the correct values that correspond to the model. Measurements from every pair of original and complementary spectral patterns are also used to obtain a pure spatial modulation for calculation of optical flow of grayscale previews” Can you clarify how the complementary patterns/measurements are used to deal with the negative values in the target $(-1, 1)$ modulation matrices during reconstruction? I see two options: i) do you pre-process the measurements, i.e. subtract the "complementary" measurements; ii) do you keep all the measurements but include the complementary patterns in the modulation matrices? An equation might help to understand these subtle but often important details.

Other minor comments on the supplementary material:

- Fig 1 S1a is too small

References:

- A. <https://dl.acm.org/doi/10.1145/3345553>
- B. <https://doi.org/10.1109/TPAMI.2021.3075228>
- C. <https://doi.org/10.1038/s41467-021-24850-x>
- D. <https://www.nature.com/articles/s41467-022-35585-8>
- E. <https://doi.org/10.1364/OE.483937>
- F. <https://doi.org/10.1364/OE.483937>

Reviewer #2 (Remarks to the Author):

This paper presents a new approach to hyperspectral video imaging based on a single-pixel detector. The imaging system utilizes a joint spatial-spectral encoding scheme, taking advantage of the compressibility of 4-dimensional hyperspectral videos compared to 2D grayscale images. By encoding the scene into highly compressed single-pixel measurements, the input data throughput is significantly reduced. Additionally, temporal correlation is obtained through optical flow analysis. The paper also introduces an optimization-based reconstruction method that retrieves high-throughput hyperspectral video from the measurements. A deep learning reconstruction technique, which accelerates the reconstruction process, is also presented. This single-pixel-based hyperspectral imager has potential applications when the storage and transmission of hyperspectral video data are difficult. Furthermore, the approach shows promise for extension to wavelengths beyond the visible domain.

Overall, this work presents an interesting contribution to the field of hyperspectral imaging. However, there are two significant issues with this manuscript that negatively impact the quality of the work. First, the short format of a communication/letter does not provide an effective format for the work presented. Second, the paper does not adequately compare the single pixel system with previous work in hyperspectral video acquisition. A detailed description of these concerns, as well as comments about the individual sections of the paper are provided below.

Significant Concerns

1) As presented, the paper is disjointed and poorly organized. The structure of the submission, consisting of a main manuscript and a significant supplement leads to a fragmented reading experience. The format does not effectively address the scope and depth of the research presented in the paper, potentially leaving readers with unanswered questions. To address these concerns, the authors should consider reorganizing the content into a larger manuscript format. This would allow for a more comprehensive and cohesive presentation of the camera setup, data acquisition, and reconstruction methods. By integrating the relevant information within the main manuscript, readers will have a clearer understanding of the research without the need for constant cross-referencing between the main text and the supplement. This will result in a more accessible and comprehensive account of the research findings.

2) The manuscript would benefit from a detailed and quantitative comparison of the authors' techniques to other works in the hyperspectral video (HSI) field in the discussion section. By including a comparison to previously published work, the authors can demonstrate the novelty of their techniques and highlight the advantages of their approach over existing methods. The manuscript should provide a thorough analysis of the strengths and weaknesses of their technique in relation to other HSI video systems. The lack of a comprehensive comparison with other works in the field raises concerns about the unique contributions of the study and the significance of the findings. Addressing these concerns and incorporating a robust comparison to relevant literature will strengthen the manuscript. The most recent references in this paper are from 2022. Several interesting papers about HSI video acquisition and deep learning approaches have been published recently. For example, Yako et al. present a novel approach for capturing hyperspectral video data in real-time. The researchers developed a camera system that utilizes a random array of Fabry-Pérot filters, which are compatible with complementary metal-oxide-semiconductor (CMOS) technology. The camera system enables video-rate acquisition of hyperspectral images. These authors propose a reconstruction algorithm that utilizes compressive sensing techniques to retrieve the hyperspectral video data from the measurements. They compare their results with conventional hyperspectral imaging techniques and show that their approach offers significant advantages in terms of data throughput and potential extension to wavelengths beyond the visible domain. In many ways this work seem similar to the work presented here. What are the advantages/disadvantages of the single pixel approach? The full reference for this paper is Yako, Motoki, et al. "Video-rate hyperspectral camera based on a CMOS-compatible random array of Fabry-Pérot filters." *Nature Photonics* 17.3 (2023): 218-223. In addition, Gutiérrez-Zaballa, et al. present a method for on-chip hyperspectral image segmentation using fully convolutional networks (FCNs) to improve scene understanding in autonomous driving. The authors propose a framework that integrates hyperspectral imaging and

deep learning techniques to accurately classify and segment objects in real-time. How do the deep learning approaches presented here compare to those used in the work by Gutiérrez-Zaballa, et al.? The full reference for this paper is Gutiérrez-Zaballa, Jon, et al. "On-chip hyperspectral image segmentation with fully convolutional networks for scene understanding in autonomous driving." *Journal of Systems Architecture* 139 (2023): 102878.

These papers are just two recent examples.

General Comments

1) In addition to the organizational challenges of the supplement, sections 2.1-2.3 seem as though they should be in Methods section, not Results. These sections describe hardware and algorithms.

2) Be clear about the time that it takes to acquire the video and then reconstruct the video. There is a data acquisition time that includes displaying the spatial and spectral encoding patterns on the DMD and running through this sequence multiple times. After this the video is reconstructed, with the deep learning algorithm showing a significant speed advantage. The benefit is the compression of the data stream that needs to be transmitted or stored. The authors mention autonomous driving, but it is not clear that this HSI system is appropriate for this type of application.

3) The figures and corresponding captions in the manuscript exhibit mediocre quality, which hampers their effectiveness in conveying information. Problems include improper labeling of panels, difficulty in reading or seeing labels, legends and boxes used for highlighting, and missing information in captions. For instance, in Figure 12, the yellow box in the second panel of the top row attracts attention but lacks explanation in the caption or accompanying text. It is only later mentioned when discussing the spectra of different regions. Similar issues arise with boxes in other panels, making them hard to discern. It is crucial to ensure that figures and captions provide clear and concise information, facilitating reader understanding of key points and context. Enhancing clarity and providing more detailed descriptions in the captions would greatly improve the overall quality and readability of the figures.

4) The paper contains several grammatical errors, particularly in relation to the usage of articles such as "the" or "a." These errors have a negative impact on the overall quality of the paper. It is important to address these grammatical issues to improve the clarity and professionalism of the writing.

5) The manuscript contains several abbreviations that are not adequately defined, especially in the sections related to the deep learning algorithm. This lack of definition poses a problem for readers trying to understand the content. It is essential to clearly define abbreviations the first time they are introduced to ensure clarity and comprehension.

6) The multimedia files are useful and informative.

7) The details of deep learning algorithms are outside my area of expertise. I reviewed these sections for clarity and did not comment on the validity of the methods employed. I did evaluate the quality of the resulting reconstructions.

Further Notes on the Main Manuscript

1. Introduction

The authors provide a good overview of existing hyperspectral imaging techniques and highlight their limitations. They emphasize the advantage of encoding both spatial and spectral information and discuss various reconstruction methods that have been explored. The paper highlights the achievement of reconstructing 128 x 128 hyperspectral video data with 64 spectral bands at speeds of 4.3 frames per second, showcasing impressive results. However, it should be noted that the reported speed does not include the video acquisition time. The authors also mention the potential applications of their technique in rovers and satellites, but it would be beneficial to further explain the significance of the increase in reconstruction speed. A more thorough explanation of this goal and its implications would enhance the reader's understanding of the paper's contributions.

2. Results

- CR is defined in the lines 236-238. It would be beneficial to readers not as familiar with compressed imaging to relate this directly to the work presented and provide consistent information about the # of patterns used for the CR values discussed in the text and shown later in the figures.

- Figure 3 is confusing. What is the line to the right connecting the single-doxel imager to the 4DTV-based recovery box represent? I thought that the gray scale video was used to determine the optical flow and then this was used as input for the full recovery algorithm. Further, this figure seems to show almost the same information as Figure 5.

- In Figure 5 the arrow at the top under the measurement sequence should be labeled as time or frame number. Figures 3 and 5 seem redundant. Each of these figures is showing the process by which the video is reconstructed. Figure 5 illustrates this process more clearly. Perhaps remove Figure 3?

- Lines 346-350. "The data windows for each frame may overlap." Exactly what does this mean? More explanation is needed here. Overall, this paragraph is not clear.

- Line 354: Define the abbreviation PDHG.

- Line 374 – The multi-resolution results are only briefly mentioned and the reader is then referred to the supplement. Why are these results relegated to just the supplement? The use of STOne patterns is an intriguing idea and provides the ability to reconstruct images at varying spatial resolutions. However, it is unfortunate that the authors do not further discuss or revisit this concept in the main body of the paper. Again, the structure of a main text + supplement may not be the most suitable choice for presenting the content of this paper, as important aspects and ideas, such as the STOne patterns, should be integrated and discussed more prominently in the main text.

- Figure 7 – The legends are impossible to read, The lines in each panel are the same color and line style making it very difficult to tell ground truth and reconstruction apart. What is meant by "good quality"?

- Line 399 – This should be Fig. 8b.

- Figure 8: Again, the legends are impossible to read and it is very difficult to tell the curves apart in each panel.

- Line 422 - "temporal correlation among 5 adjacent frames" How much time is there between the frames?

- Line 423 - Define CNN

- Lines 449 – 471 Multiple articles are missing - "the" and "a"

- Table 1 shows PSNR/SSIM/SAM. This seems to be mislabeled in the table caption. Also, they should discuss the significance of the numbers in the table. Are these numbers good? How do we know? What do they tell us? This is the quantitative analysis of their reconstruction and they need to explain and expound on the numbers presented in Table 1.

- Quantitative measures of reconstruction are presented for the deep learning section in Table 1. Why is this type of quantitative analysis not presented for the iterative algorithm? This seems like a significant omission. It would have been insightful to compare and discuss the results of the 2 different reconstruction methods, beyond reconstruction time.

- Figure 11: The results shown in this figure should be described in more detail. For example, explain why different CR values yield different results.

- Line 540 - What is meant by "reasonable match"?

- Figure 12: Define what the boxes are in the top row. The definition comes later, but it should be here also. It is confusing to see these in the panels and to not know what they are. I thought there was a problem with the figure when I first looked at it. Also, the boxes are extremely difficult to see in the other panels on the top row.

- Line 557 : "Significant gains in reconstruction time." This is true, but a full discussion about why this is important is missing. What application would benefit from this ability? What are the limitations in speed of the technique? Acquisition? How do you speed that up? How do you decide what CR is appropriate?

- Figure 13: The legends are difficult to read.

3. Discussion

This section is missing context. This is by far the weakest section of the paper. To understand the significance of the work presented here, the authors need to compare their work quantitatively to previous work.

4. Methods

- Section 4.2 is poorly written. There are multiple undefined abbreviations and the content is very difficult to follow.

- Line 652 – Replace don't with do not in formal writing.

- Line 674 This statement should have been included when presenting the metrics in Table 1. In fact, this whole section on Metrics would help with the explanation of this table.

Notes on the Supplement

- This section provides important details that are missing from the main manuscript.

Section 1 - Calibration

- It is impossible to read the labels in Figure S1a. A larger font and perhaps a different color (e.g., white) for the labels would be helpful.

- They mention slit width and its obvious impact on spectral resolution but they do not mention what slit width was used in this setup. What slit width was used in this system to provide the quoted resolution? Why was this slit width chosen?

- Line 31: Define "complementary and calibration patterns."

Section 2 – STOne patterns

- Fig. S5 - Label the panels. In the caption and text it says that panel a is grayscale, but the figure seems to show RGB. In panel b the outline around the frame used is difficult to see.

Section 3

- Line 86 Redefine "complementary pattern pair" to be clear.

- Line 94 – Describe the saddle point problem in 1-2 brief sentences. Do not just reference the other paper (reference 2). The reader should be able to have a general understanding of the main idea of the saddle point problem from reading this paper and can read the reference for the details.

- Line 97: This is the same issue as in Line 94. Provide a brief summary of the algorithm and then reference the reader to the other paper for the details.

- Line 97 – What are the properties of the Gaussian used?
- Fig S6. When the reference to the other paper’s color code is made in the text, mention that it is also shown in Figure S6.
- Line 113-14: Tell the reader about the method briefly, do not just the reference the other work.
- Lines 112-126 – This is a very abrupt transition. This is an important paragraph pointing out what is needed to achieve high CR and fast reconstruction. Tie this back to findings of the main paper.
- Elaborate on the description of Figure S8.

Section 4

- Line 144: define/remind the reader what is meant by “test patch.”
- Lines 140-145: There are many missing articles (the, a, etc.) in this section.
- Lines 160-162 are repetitive.
- Figure S10 This figure needs a better caption. The panels need to be labeled (and perhaps shown side by side rather than in a vertical stack).
- Line 212 - Exactly how and why were the Hadamard patterns used? This is not clear from this part of the text. In the main manuscript it is mentioned that these patterns are used for the spectral encoding. Why? I presume this is because there may not be a need to reconstruct at different resolutions in the spectral dimension?

Reviewer #3 (Remarks to the Author):

This paper proposes a compact imaging system for hyperspectral video acquisition, which is a first-of-its-kind of the single-pixel-based hyperspectral imager. The author designs optical flow assisted 4DTV regularization and DNN for reconstruction, which achieves 4 frames per second for a $128 \times 128 \times 64$ hyperspectral image. The results of simulation data and real-world data demonstrate the feasibility of this imaging system and promising reconstruction quality. I think this paper cannot meet the publication criteria set by this journal.

1. The DNN network introduced in this paper seems to exhibit a level of innovation that is relatively modest, resulting in a somewhat conventional approach. Further exploration and integration of more advanced techniques could potentially enhance the originality and uniqueness of the proposed network architecture.
2. The methodology introduced in this paper presents a challenge in terms of achieving a substantial spectral data flux. The data flux of $128 \times 128 \times 64 \times 4$ may not fully align with high throughput expectations. Additionally, it is notable that both spatial and temporal resolutions appear to be on the lower side, possibly suggesting room for improvement in capturing finer details and temporal dynamics. Addressing these limitations could contribute to enhancing the comprehensiveness and accuracy of the proposed approach.
3. The assertion made in line 152 regarding a "900x data throughput" is not without potential ambiguities. It is important to acknowledge that utilizing a DMD-based single-pixel imaging system itself holds the capability to achieve elevated temporal or spatial resolutions. The system proposed in this paper, however, appears to trade off temporal and spatial resolution in favor of spectral resolution. The claim of "900x data throughput" warrants further scrutiny and clarification, taking into account the nuanced trade-offs between these different aspects of imaging performance.
4. The comparative advantages of this system in relation to other hyperspectral video acquisition systems are not clearly apparent in the paper's presentation. Further comprehensive discussion and substantiation are required to elucidate and establish these potential differentiators.

5. The imaging model of the optical modulation in Fig. 1 is unclear. The imaging model is crucial for the reconstruction as the data fidelity term. Particularly, the purpose of the slit and the diffraction grating should be described and analyzed. The focal plane of each lens should be indicated.

Response to Reviewers' Comments

We thank all the reviewers for the time and effort they put into reading the manuscript and for the valuable and constructive comments. The track changes are kept on in the revised manuscript. The referenced line numbers are with respect to the manuscript with track changes on. In this file, each original comment from the reviewer is listed followed by the author's response to the comment.

Reviewer #1 (Remarks to the Author):

This study proposes and demonstrates the use of single pixel imaging for video rate hyperspectral imaging. To do this, the authors present an optical design that enables joint spatial-spectral encoding using a single digital micromirror device and a single point detector, where traditional single-pixel approaches rely on either spatial or spectral encoding or multiple detection arms. I found the paper very interesting and compelling. The proposed encoding scheme is clever, as spatial-spectral encoding allows better exploitation of the compressibility of temporal hypercubes, thus reducing the number of measurements and allowing faster acquisition. To the best of my knowledge, this approach is new and is likely to influence the field of single-pixel imaging and, more generally, computational imaging. Other notable results include the use of an optical flow assisted reconstruction algorithm that estimates the optical flow from the raw measurements. The authors convincingly demonstrate their claims and conclusions by imaging a rotating colour wheel at more than 4 fps (128 x 128 x 64 hypercube with a spectral resolution of ~7 nm).

Comments on the main document:

- I strongly recommend discussing references A, B, C, D and E in the introduction and/or discussion. Ref A proposes an adaptive method that can drastically reduce the number of measurements required for hyperspectral imaging. Ref. B proposed another adaptive method that allows hyperspectral video with a spatial resolution of 600 x 900 pixels, a spectral resolution of 10 nm over visible wavebands, and a frame rate of 18 fps. Both methods rely on similar but different reconstruction assumptions and hardware. Refs. C and D introduced fast alternatives to DMDs for single pixel modulation. Although not specifically designed for spectral imaging, they could be extended to spectral imaging. In particular, it would be interesting to discuss whether and how such modulation strategies could be combined with the proposed methodology. Reference F describes an ecosystem for "standard" single pixel hyperspectral imaging, where 128 x 128 x 2048 hypercube are acquired in 12s. It also presents a database of more than 150 hypercubes.
Author: Thank you for suggesting these relevant literatures. We have discussed all of them plus some other papers in Line 194-191 as part of the related work in *Introduction* section, where we have provided the approach and the key results for each paper. In the Discussion section in Line 775-811, we have added the advantages and novelty of the proposed single-pixel approach for hyperspectral video imaging over existing methods reported in the literature. Further, we added discussion of increasing the resolution of the SDI in Line 812-858. We have also discussed whether and how the fast pattern modulation strategies could be combined with the proposed methodology as you suggested in the Line 848-858.
- Line 75: "Single-pixel imaging is an emerging computational imaging technique". I would say that single-pixel imaging is now an established technique.
Author: Thank you. We agree and have deleted "emerging".
- Line 123: "Soldevila et al. reported a compression ratio of 3000:1". Compression ratios often carry little information in the sense that the image dimension (i.e., number of pixels) used for reconstruction can be chosen arbitrarily. For a given number of measurements, e.g. $M = 1,024$, the image can be

reconstructed by choosing $N = 4,096$ pixels, resulting in a compression ratio of 4:1, or by choosing $N = 16,384$ pixels, resulting in a compression ratio of 16:1. Therefore, two reconstructions with the same spatial resolution (in terms of the ability to distinguish between two nearby objects) can have arbitrarily different compression ratios.

Author: Thanks for pointing out. We have edited the related sentences with more details added and analyzed the results of this paper in Line 137-149.

- Line 208: of “?? spatial patterns and ?? spectral”
Author: Thanks for pointing out. The pdf version has format issue. It should be “ N spatial patterns and K spectral patterns” as corrected in Line 275.
- Line 230: "In practice, the data windows for each frame may overlap or have different widths." I think this sentence needs clarification. Does this "width" refer to the "sliding window" mentioned in line 349? As I am probably missing something here, I do not see the interest in introducing equation (2), which brings no additional information. Unless you just want to define b and R for equation 3?
Author: Yes, equation (2) needs to be introduced to define the 4DTV regularization in equation (3). Thanks for pointing out. We have rewritten this sentence in Line 295-306 with clearer explanations of the data window.
- Line 250: "Both are binary patterns that can be easily implemented on the DMD". I would refer to the section in the supplementary document where you explain how to handle the negative value, as there are several variants.
Author: Details of implementation of the binary patterns are added in “Complementary patterns” in *Methods* section.
- Equation (3). To reconstruct the dynamic hypercube (x,y,λ,t) , the authors consider a 4DTV prior (isotropic TV in space, anisotropic TV over the temporal and spectral dimensions). The spectral prior favours piecewise linear spectra, while the temporal prior favours piecewise constant dynamics. Could you motivate such a prior? Why not just enforce smooth variations across the temporal and spectral dimensions? This should be less computationally demanding. The same question applies to the L1-based optical flow prior, which allows for localised motion errors.
Author: As added in Line 366-370, the temporal piecewise constant approximation is a natural extension of the spatial piecewise constant property. For the spectra in the visible and the near infrared for most natural scenes, piecewise constant would obviously be an inaccurate approximation, whereas piecewise linear is a good enough approximation. For spectra in other bands, other priors may be used. The temporal TV term in the 4DTV algorithm includes optical flow between any two frames with a temporal distance closer than or equal to 2 for better regularization.
- Line 310: “However, the calculation of optical flow requires the knowledge of video frames while, on the other hand, the optical flow is expected to be used in video reconstruction for improved results”. I would recommend mentioning this point earlier, typically right after equation (3), which introduces v_x and v_y , and explaining that a method for estimating the vector field from the raw measurement will be introduced later. I believe that the current description of this "trick" (lines 313-329) is not clear enough, in contrast to section 4.1 and sections 1 and 2 of the supplementary document, which describe very well what is going on. I would suggest deleting lines 313-329 (and Figure 3) and referring the reader to Section 4.1 and the Supplementary Document. An alternative would be to improve the paragraph, where it may be more important to focus on the idea of complementary patterns giving access to the greyscale video, rather than the idea of "preview" (which could be removed for simplicity?).
Author: Thanks for the valuable suggestion. We agree and have followed the suggestions and mentioned the point earlier in Line 383-394 and deleted lines 313-329 referring supplementary document, which makes the manuscript more coherent and concise.

- Line 347: “In practice the data windows for each frame may overlap, or be of different widths”. As mentioned above, I have not been able to understand what the “width” refers to and how

Author: Thanks again for pointing out. The original sentence was not clearly written. The sentence has been rewritten with clearer explanations in Line 295-306.
- Line 420: How does the reconstruction approach compares to the one described in Ref. F? What are the main differences?

Author: Details on Ref. F are discussed in Line 178-191. In short, Ref. F uses a commercial spectrometer as the detector with a 2D array sensor for spectral resolvability whose spectral resolution directly decides the spectral resolution of the hypercube. Therefore, the detector is not true “single pixel”. Additionally, Ref. F only employs spatial compression and not spectral compression. Also, although some type of spatial compression happened during sampling, no compressive sensing (CS) methods are used in the system for sampling or reconstruction, which leads to long sampling time and relatively low reconstruction quality. The highest compression ratio demonstrated is 32:1.
- Line 430: “First, the grayscale preview video of 5 frames is reconstructed from spectrally complementary measurements”

Author: The comment did not mention what is the issue with this sentence. But regarding complementary patterns, details of implementation are added in “Complementary patterns” in *Methods* section.
- Line 523: “to GPU memory limit, the models are designed to recover hyperspectral image patches of spatial size 32×32 with 41 spectral”. Where is the memory bottleneck of the proposed deep learning approach? How many learnable parameters do you have in the “temporal” module? In the “hyperspectral” module?

Author: Thanks for bringing it up. The related content has been added in Line 861-866 in *Discussion*.
- As shown in Fig. S9, processing small patches leads to discontinuities at the boundary of the patches. How do you correct for these artefacts? Have you considered the reconstruction of “moving” patches that you can merge using another (small) CNN?

Authors: Thanks for bringing it up. There could be multiple approaches to smooth the boundary of patches, including reconstructing overlapping patches which are then averaged in overlap area, applying smoothing filters to the boundary, using another CNN for smoothing as you suggested, etc. The main goal of the proposed deep learning method is to validate possibility of acceleration using neural networks. Also, we intentionally did not try to smooth the boundary in order to demonstrate the raw reconstruction results of the network design. Smoothing boundaries could be a topic for the future papers which focus on algorithm optimization for the SDI.
- Line 628. “After the measurements, calibration values are calculated from these complementary patterns, and are subtracted from the measurement outputs to get the correct values that correspond to the model. Measurements from every pair of original and complementary spectral patterns are also used to obtain a pure spatial modulation for calculation of optical flow of grayscale previews” Can you clarify how the complementary patterns/measurements are used to deal with the negative values in the target (-1, 1) modulation matrices during reconstruction? I see two options: i) do you pre-process the measurements, i.e., subtract the “complementary” measurements; ii) do you keep all the measurements but include the complementary patterns in the modulation matrices? An equation might help to understand these subtle but often important details.

Author: Thank for bringing it up. Details of how to use complementary patterns are added in “Complementary patterns” in *Methods* section. It uses pre-processing of the measurements.

Other minor comments on the supplementary material

- Fig 1 S1a is too small

Author: we have made Fig S1a larger and more readable

References:

- A. <https://dl.acm.org/doi/10.1145/3345553>
- B. <https://doi.org/10.1109/TPAMI.2021.3075228>
- C. <https://doi.org/10.1038/s41467-021-24850-x>
- D. <https://www.nature.com/articles/s41467-022-35585-8>
- E. <https://doi.org/10.1364/OE.483937>
- F. <https://doi.org/10.1364/OE.483937>

Reviewer #2 (Remarks to the Author):

This paper presents a new approach to hyperspectral video imaging based on a single-pixel detector. The imaging system utilizes a joint spatial-spectral encoding scheme, taking advantage of the compressibility of 4-dimensional hyperspectral videos compared to 2D grayscale images. By encoding the scene into highly compressed single-pixel measurements, the input data throughput is significantly reduced. Additionally, temporal correlation is obtained through optical flow analysis. The paper also introduces an optimization-based reconstruction method that retrieves high-throughput hyperspectral video from the measurements. A deep learning reconstruction technique, which accelerates the reconstruction process, is also presented. This single-pixel-based hyperspectral imager has potential applications when the storage and transmission of hyperspectral video data are difficult. Furthermore, the approach shows promise for extension to wavelengths beyond the visible domain.

Overall, this work presents an interesting contribution to the field of hyperspectral imaging. However, there are two significant issues with this manuscript that negatively impact the quality of the work. First, the short format of a communication/letter does not provide an effective format for the work presented. Second, the paper does not adequately compare the single pixel system with previous work in hyperspectral video acquisition. A detailed description of these concerns, as well as comments about the individual sections of the paper are provided below.

Significant Concerns

1) As presented, the paper is disjointed and poorly organized. The structure of the submission, consisting of a main manuscript and a significant supplement leads to a fragmented reading experience. The format does not effectively address the scope and depth of the research presented in the paper, potentially leaving readers with unanswered questions. To address these concerns, the authors should consider reorganizing the content into a larger manuscript format. This would allow for a more comprehensive and cohesive presentation of the camera setup, data acquisition, and reconstruction methods. By integrating the relevant information within the main manuscript, readers will have a clearer understanding of the research without the need for constant cross-referencing between the main text and the supplement. This will result in a more accessible and comprehensive account of the research findings.

Author: Thank you for the suggestion. As elaborated in the answers later, we have integrated relevant content for camera setup, data acquisition, and reconstruction methods as well as the multi-resolution reconstruction using STOne patterns from the Supplement material into the main paper for coherent

presentation.

2) The manuscript would benefit from a detailed and quantitative comparison of the authors' techniques to other works in the hyperspectral video (HSI) field in the discussion section. By including a comparison to previously published work, the authors can demonstrate the novelty of their techniques and highlight the advantages of their approach over existing methods. The manuscript should provide a thorough analysis of the strengths and weaknesses of their technique in relation to other HSI video systems. The lack of a comprehensive comparison with other works in the field raises concerns about the unique contributions of the study and the significance of the findings. Addressing these concerns and incorporating a robust comparison to relevant literature will strengthen the manuscript. The most recent references in this paper are from 2022. Several interesting papers about HSI video acquisition and deep learning approaches have been published recently. For example, Yako et al. present a novel approach for capturing hyperspectral video data in real-time. The researchers developed a camera system that utilizes a random array of Fabry–Pérot filters, which are compatible with complementary metal-oxide-semiconductor (CMOS) technology. The camera system enables video-rate acquisition of hyperspectral images. These authors propose a reconstruction algorithm that utilizes compressive sensing techniques to retrieve the hyperspectral video data from the measurements. They compare their results with conventional hyperspectral imaging techniques and show that their approach offers significant advantages in terms of data throughput and potential extension to wavelengths beyond the visible domain. In many ways this work seem similar to the work presented here. What are the advantages/disadvantages of the single pixel approach? The full reference for this paper is

Yako, Motoki, et al. "Video-rate hyperspectral camera based on a CMOS-compatible random array of Fabry–Pérot filters." *Nature Photonics* 17.3 (2023): 218-223.

In addition, Gutiérrez-Zaballa, et al. present a method for on-chip hyperspectral image segmentation using fully convolutional networks (FCNs) to improve scene understanding in autonomous driving. The authors propose a framework that integrates hyperspectral imaging and deep learning techniques to accurately classify and segment objects in real-time. How do the deep learning approaches presented here compare to those used in the work by Gutiérrez-Zaballa, et al.? The full reference for this paper is Gutiérrez-Zaballa, Jon, et al. "On-chip hyperspectral image segmentation with fully convolutional networks for scene understanding in autonomous driving." *Journal of Systems Architecture* 139 (2023): 102878.

These papers are just two recent examples.

Author: Thank you for suggesting the papers to make the manuscript more thorough and convincing. We have discussed many more recent literatures including the two you suggested as related work in the *Introduction* section in Line 194-191, presenting the approach, reported resolution, compression ratio, etc.

In the *Discussion* section, we have added a detailed and thorough analysis on our approach compared to existing methods in literature. The advantages of using a single-pixel detector vs. 2D array detector are presented in Line 775-787. The motivation and novelty of the achieved high compression ratio of 900:1 are discussed in Line 787-811. The analysis on further increasing resolution is presented in Line 812-858.

In particular, the first paper you suggested by Yako, Motoki, et al. is presented in Line 114-122. In short, though the paper demonstrates an impressive result of hyperspectral imaging at 32.3 fps at VGA resolution, it reconstructs only 20 spectral bands. It uses a 2D array sensor and the sampling compression ratio is limited to 20:1. It doesn't enjoy the benefits of single-pixel based approach and a high compression ratio. The second paper you suggested by Gutiérrez-Zaballa, et al. is presented in Line 174-178. In this paper, the task is object segmentation and classification from a full hypercube. The authors used a network design adapted from the widely used U-Net structure which was original intended for and has been successfully applied in accurate image segmentation. In our paper, the task of deep learning is hyperspectral video

reconstruction from highly compressed random measurements. We design the neural network structure following the strategy that signal sparsity in 4D space could be utilized to its full extent.

General Comments

1) In addition to the organizational challenges of the supplement, sections 2.1-2.3 seem as though they should be in Methods section, not Results. These sections describe hardware and algorithms.

Author: We debated before whether to put sections 2.1-2.3 in Methods, but we eventually feel hardware and algorithms can also be regarded as important results of our contribution. We looked at papers published in Nature Communications on similar topics and found some of them also put hardware and algorithm in results. We decided similarly.

2) Be clear about the time that it takes to acquire the video and then reconstruct the video. There is a data acquisition time that includes displaying the spatial and spectral encoding patterns on the DMD and running through this sequence multiple times. After this the video is reconstructed, with the deep learning algorithm showing a significant speed advantage. The benefit is the compression of the data stream that needs to be transmitted or stored. The authors mention autonomous driving, but it is not clear that this HSI system is appropriate for this type of application.

Author: Thanks for help clarifying. The reported 4.3 fps has considered both data acquisition time and reconstruction time. Here, using deep learning, reconstruction is faster than data acquisition. Therefore, the frame rate is constraint by the data acquisition time which is 4.3 fps when we operate the DMD at 5 kHz. The reconstruction can be performed simultaneously and in real-time with data acquisition.

In the experiment, we did not push the data acquisition speed to its limit by operating the DMD pattern rate at 22kHz because we are not aiming for only increasing frame rate but also would like to retain high reconstruction quality. The system can operate at higher frame rate, as discussed in Line 831-858, while maintaining the reconstruction quality with more optimized deep learning algorithms, e.g. when training the neural network with the corresponding noise level, or using another denoising network.

Autonomous driving is mentioned at the beginning of *Introduction* section when talking about applications for high resolution hyperspectral video imaging in the broad sense. Autonomous driving should be a valid application for hyperspectral video imaging in general. As for the SDI system, because it does not need a 2D detector array which pushbroom, CASSI, and other snapshot systems require, its capability to easily extend to infrared imaging and ultra-low light imaging endows it with the potential for autonomous driving.

3) The figures and corresponding captions in the manuscript exhibit mediocre quality, which hampers their effectiveness in conveying information. Problems include improper labeling of panels, difficulty in reading or seeing labels, legends and boxes used for highlighting, and missing information in captions. For instance, in Figure 12, the yellow box in the second panel of the top row attracts attention but lacks explanation in the caption or accompanying text. It is only later mentioned when discussing the spectra of different regions. Similar issues arise with boxes in other panels, making them hard to discern. It is crucial to ensure that figures and captions provide clear and concise information, facilitating reader understanding of key points and context. Enhancing clarity and providing more detailed descriptions in the captions would greatly improve the overall quality and readability of the figures.

Author: Thanks for the suggestion. We have merged Fig.12 and Fig. 13 as Fig. 12 and rewritten the caption to provide clear information. We have also improved some other figures and captions.

4) The paper contains several grammatical errors, particularly in relation to the usage of articles such as "the" or "a." These errors have a negative impact on the overall quality of the paper. It is important to address these grammatical issues to improve the clarity and professionalism of the writing.

Author: Thank you for your careful reading of the paper. We have improved the grammar.

5) The manuscript contains several abbreviations that are not adequately defined, especially in the sections related to the deep learning algorithm. This lack of definition poses a problem for readers trying to understand the content. It is essential to clearly define abbreviations the first time they are introduced to ensure clarity and comprehension.

Author: We have defined all the abbreviations.

6) The multimedia files are useful and informative.

7) The details of deep learning algorithms are outside my area of expertise. I reviewed these sections for clarity and did not comment on the validity of the methods employed. I did evaluate the quality of the resulting reconstructions.

Further Notes on the Main Manuscript

1. Introduction

The authors provide a good overview of existing hyperspectral imaging techniques and highlight their limitations. They emphasize the advantage of encoding both spatial and spectral information and discuss various reconstruction methods that have been explored. The paper highlights the achievement of reconstructing 128 x 128 hyperspectral video data with 64 spectral bands at speeds of 4.3 frames per second, showcasing impressive results. However, it should be noted that the reported speed does not include the video acquisition time. The authors also mention the potential applications of their technique in rovers and satellites, but it would be beneficial to further explain the significance of the increase in reconstruction speed. A more thorough explanation of this goal and its implications would enhance the reader's understanding of the paper's contributions.

Author: Thanks for help clarifying. As explained previously, the reported 4.3 fps is for real-time imaging when reconstruction is faster than or equal to data acquisition speed. The SDI system easily allows for higher frame rate.

In the *Discussion* section, we have added a detailed and thorough analysis on our approach compared to existing methods in literature. The advantages of using a single-pixel detector vs. 2D array detector are presented in Line 775-787. The motivation and novelty of the achieved high compression ratio of 900:1 are discussed in Line 787-811. The analysis on further increasing resolution is presented in Line 812-858.

2. Results

•CR is defined in the lines 236-238. It would be beneficial to readers not as familiar with compressed imaging to relate this directly to the work presented and provide consistent information about the # of patterns used for the CR values discussed in the text and shown later in the figures.

Author: Thanks for bringing it up. The number of measurements is provided in section 2.4 Line 445 for the 4DTV-based recovery. They are provided in Supplement material in Line 169-173 for simulation data and in Line 240-243 for the experimental data of deep learning approach.

•Figure 3 is confusing. What is the line to the right connecting the single-doxel imager to the 4DTV-based recovery box represent? I thought that the gray scale video was used to determine the optical flow and then this was used as input for the full recovery algorithm. Further, this figure seems to show almost the same information as Figure 5.

Author: The line represents the data flow of raw single pixel measurements. The raw measurements are used in two pathways. On one hand, they are for reconstruction of grayscale video. On the other hand,

they are used, together with the reconstructed grayscale video, for final hyperspectral reconstruction. We agree Figure 5 is good enough so have removed Figure 3.

- In Figure 5 the arrow at the top under the measurement sequence should be labeled as time or frame number. Figures 3 and 5 seem redundant. Each of these figures is showing the process by which the video is reconstructed. Figure 5 illustrates this process more clearly. Perhaps remove Figure 3?

Author: We have labeled the arrow with “Time”. Thanks for pointing out. We agree Figure 5 is good enough so have removed Figure 3.

- Lines 346-350. “The data windows for each frame may overlap.” Exactly what does this mean? More explanation is needed here. Overall, this paragraph is not clear.

Author: We have rewritten related sentences to explain the “data window” in Line 295-306 with clearer explanations of the data window.

- Line 354: Define the abbreviation PDHG.

Author: We have defined it in Line 482.

- Line 374 – The multi-resolution results are only briefly mentions and the reader is then referred to the supplement. Why are these results relegated to just the supplement? The use of STOne patterns is an intriguing idea and provides the ability to reconstruct images at varying spatial resolutions. However, it is unfortunate that the authors do not further discuss or revisit this concept in the main body of the paper. Again, the structure of a main text + supplement may not be the most suitable choice for presenting the content of this paper, as important aspects and ideas, such as the STOne patterns, should be integrated and discussed more prominently in the main text.

Author: Thanks for the suggestion. The related discussion on STOne patterns are integrated in the main paper in Section 2.3 Line 427-434 and Section 2.4 Line 558-570.

- Figure 7 – The legends are impossible to read, The lines in each panel are the same color and line style making it very difficult to tell ground truth and reconstruction apart. What is meant by “good quality”?

Author: Thank you for pointing out. We have made the legends larger. In each panel, the two lines are the same color but of a solid style and a dashed style to differentiate them. We have deleted the ambiguous wording ‘good spectral match’.

- Line 399 – This should be Fig. 8b.

Author: Thank you. We have changed it.

- Figure 8: Again, the legends are impossible to read and it is very difficult to tell the curves apart in each panel.

Author: We have made Figure 8 larger and more readable.

- Line 422 - “temporal correlation among 5 adjacent frames” How much time is these between the frames?

Author: Thank you for bringing it up. Since this is for simulation data, we generated each video sequence of 5 frames by spatially translating a hyperspectral image so it doesn’t really involve the time difference here. The neural network learns the temporal correlation from the differences in the 5 frames of all the training data.

•Line 423 - Define CNN

Author: Thank you. We have defined it.

•Lines 449 – 471 Multiple articles are missing - “the” and “a”

Author: Thank you. We have changed it.

•Table 1 shows PSNR/SSIM/SAM. This seems to be mislabeled in the table caption. Also, they should discuss the significance of the numbers in the table. Are these numbers good? How do we know? What do they tell us? This is the quantitative analysis of their reconstruction and they need to explain and expound of the numbers presented in Table 1.

Author: Thank you. It was indeed mislabeled and we have changed it to PSNR/SSIM/SAM. We have discussed the significance of the numbers here in Line 653-656 and have deleted the “Metrics” section in Methods.

•Quantitative measures of reconstruction are presented for the deep learning section in Table 1. Why is this type of quantitative analysis not presented for the iterative algorithm? This seems like a significant omission. It would have been insightful to compare and discuss the results of the 2 different reconstruction methods, beyond reconstruction time.

Author: Thanks for bringing it up. SSIM/PSNR/SAM are objective metrics typically useful as a coarse measure of the average objective image quality for a large dataset where it is impossible to display all the images to readers. However, such objective metrics do not fully align with human perception and are more probably to deviate a lot from human perception when used for a single image. So human perception is the “ground truth” measure and the objective metric numbers are a “convenient” way to get close to human perception when averaged across many images so the deviations may cancel out. We have demonstrated the results of the iterative algorithm to readers so have not provided the metric numbers.

•Figure 11: The results shown in this figure should be described in more detail. For example, explain why different CR values yield different results.

Author: Thank you. We have added the analysis in Line 676-678.

•Line 540 - What is meant by “reasonable match”?

Author: Thank you. The sentence is poorly written. We have rewritten it in Line 713-716.

•Figure 12: Define what the boxes are in the top row. The definition comes later, but it should be here also. It is confusing to see these in the panels and to not know what they are. I thought there was a problem with the figure when I first looked at it. Also, the boxes are extremely difficult to see in the other panels on the top row.

Author: Thank you. We have merged Fig.12 and Fig. 13 and explain the boxes in the caption. The boxes are also redrawn.

•Line 557 : “Significant gains in reconstruction time.” This is true, but a full discussion about why this is important is missing. What application would benefit from this ability? What are the limitations in speed of the technique? Acquisition? How do you speed that up? How do you decide what CR is appropriate?

Author: Thank you for valuable suggestion. We have added the content in Line 736-748

•Figure 13: The legends are difficult to read.

Author: We have made Figure 13 larger and more readable (now Figure 11 (b)).

3. Discussion

This section is missing context. This is by far the weakest section of the paper. To understand the significance of the work presented here, the authors need to compare their work quantitatively to previous work.

Author: Thank you for the suggestion. As mentioned before, in the Discussion section, we have added a detailed and thorough analysis on our approach compared to existing methods in literature, including the advantages of using a single-pixel detector vs. 2D array detector, the motivation and novelty of the achieved high compression ratio of 900:1, and the analysis on further increasing resolution, etc.

4. Methods

•Section 4.2 is poorly written. There are multiple undefined abbreviations and the content is very difficult to follow.

Author: Thank you. We have defined the abbreviations and made certain structural changes. This section may require some background knowledge on deep learning to fully follow. We would like to explain more clearly but, limited by the length of the article, we eventually decide not to include too much explanation which may be acquired from literature.

•Line 652 – Replace don't with do not in formal writing.

Author: Thank you. We have changed it.

•Line 674 This statement should have been included when presenting the metrics in Table 1. In fact, this whole section on Metrics would help with the explanation of this table.

Author: We have moved Metrics section to where Table 1 is presented in the main text.

Notes on the Supplement

•This section provides important details that are missing from the main manuscript.

Section 1 - Calibration

•It is impossible to read the labels in Figure S1a. A larger font and perhaps a different color (e.g., white) for the labels would be helpful.

Author: Thanks for the suggestion. We have made Fig. S1a larger and readable. We tried other colors for the font and found the current color still works best.

•They mention slit width and its obvious impact on spectral resolution but they do not mention what slit width was used in this setup. What slit width was used in this system to provide the quoted resolution? Why was this slit width chosen?

Author: Thanks for pointing out. The slit width and design factors are added in Line 23 -26.

•Line 31: Define “complementary and calibration patterns.”

Author: Complementary patterns are defined in *Methods* section in main paper. “Calibration patterns” is deleted from the sentence because they are the same as the complementary patterns.

Section 2 – STOne patterns

•Fig. S5 - Label the panels. In the caption and text it says that panel a is grayscale, but the figure seems to show RGB. In panel b the outline around the frame used is difficult to see.

Author: Thank you. We have changed the typo “grayscale” to “hyperspectral”. We have labeled the panels and changed the color of the outline.

Section

3

•Line 86 Redefine “complementary pattern pair” to be clear.

Author: Complementary patterns are defined in Methods section in main paper. Due to the length of that section, we have referred readers to the main paper.

•Line 94 – Describe the saddle point problem in 1-2 brief sentences. Do not just reference the other paper (reference 2). The reader should be able to have a general understanding of the main idea of the saddle point problem from reading this paper and can read the reference for the details.

Author: Thanks for mentioning this point. We did not express clearly. We did not formulate Equation S1 into the saddle point problem by ourselves. Ref. 2 provides the solver for solving the linear inverse problem for Eq. S1 which we used in the paper. Ref. 2 deals with a more general and more complicated optimization problem which it calls “saddle point problem” in order to include many related mathematical problems under a unified and larger context. Ref. 2 explains how the linear inverse problem can be formulated and solved under the larger context of the saddle point problem from a mathematical point of view. So, the saddle point problem should not be the focus here. Currently there are many mature solvers for linear inverse problems. We are only using one of them. To avoid confusion, we have deleted the “saddle point problem” here in order not to lead reader to an irrelevant direction.

•Line 97: This is the same issue as in Line 94. Provide a brief summary of the algorithm and then reference the reader to the other paper for the details.

Author: Thank you for the suggestion. We added a summary of the approach in Line 104 -108.

•Line 97 – What are the properties of the Gaussian used?

Author: The properties of the Gaussian is added in Line 108 -109.

•Fig S6. When the reference to the other paper’s color code is made in the text, mention that it is also shown in Figure S6.

Author: Thank you. We have mentioned that.

•Line 113-14: Tell the reader about the method briefly, do not just the reference the other work.

Author: Like mentioned in our answer above, to avoid confusion, we have deleted the “saddle point problem” here in order not to lead reader to an irrelevant direction. The code provided by the referenced paper can be directly used for solving typical linear inverse problems such as Equation (3). We did not formulate it into a saddle point problem by ourselves.

•Lines 112-126 – This is a very abrupt transition. This is an important paragraph pointing out what is needed to achieve high CR and fast reconstruction. Tie this back to findings of the main paper.

Author: Thank you for pointing out. We have moved this part to the main paper in Line 477- 499.

•Elaborate on the description of Figure S8.

Author: We have added more description in Line 487-496 in the main paper.

Section 4

- Line 144: define/remind the reader what is meant by “test patch.”

Author: We have edited in Line 157-158.

- Lines 140-145: There are many missing articles (the, a, etc.) in this section.

Author: we have added the missing articles.

- Lines 160-162 are repetitive.

Author: Thank you. We have deleted the repetitive sentence.

- Figure S10 This figure needs a better caption. The panels need to be labeled (and perhaps shown side by side rather than in a vertical stack).

Author: Thank you. We have labeled the panels and improved caption.

- Line 212 - Exactly how and why were the Hadamard patterns used? This is not clear from this part of the text. In the main manuscript it is mentioned that these patterns are used for the spectral encoding. Why? I presume this is because there may not be a need to reconstruct at different resolutions in the spectral dimension?

Author: Thanks for pointing out. We have added more explanations in Line 234-236. Yes, because we do not aim for multi-resolution spectral reconstruction, the permuted Hadamard patterns are used to provide the randomness required by CS theory.

Reviewer #3 (Remarks to the Author):

This paper proposes a compact imaging system for hyperspectral video acquisition, which is a first-of-its-kind of the single-pixel-based hyperspectral imager. The author designs optical flow assisted 4DTV regularization and DNN for reconstruction, which achieves 4 frames per second for a 128x128x64 hyperspectral image. The results of simulation data and real-world data demonstrate the feasibility of this imaging system and promising reconstruction quality. I think this paper cannot meet the publication criteria set by this journal.

1. The DNN network introduced in this paper seems to exhibit a level of innovation that is relatively modest, resulting in a somewhat conventional approach. Further exploration and integration of more advanced techniques could potentially enhance the originality and uniqueness of the proposed network architecture.

Author: Thanks for the suggestion. Since the proposed SDI encoding and recovery framework are new, we are the first to solve the particular reconstruction problem of the SDI. Together with proposing the optical hardware, encoding framework, and optimization algorithm in the paper, our goal regarding proposing the deep learning method is to explore and verify the feasibility of using DNN to greatly accelerate reconstruction. We have demonstrated fast reconstruction and promising quality using the DNN following the strategy to integrate and exploit signal correlation in 4D space. Based on this initial success, we are confident that there could be potentially many more different types of deep learning methods that can further improve performance, e.g., the unrolling network which unrolls the iterative optimization solver into a feed-forward network structure, the deep image prior-based algorithms, generative neural networks, adversarial training, etc. We plan to keep exploring the more optimized and advanced techniques as one of the future research directions.

2. The methodology introduced in this paper presents a challenge in terms of achieving a substantial spectral data flux. The data flux of $128 \times 128 \times 64 \times 4$ may not fully align with high throughput expectations. Additionally, it is notable that both spatial and temporal resolutions appear to be on the lower side, possibly suggesting room for improvement in capturing finer details and temporal dynamics. Addressing these limitations could contribute to enhancing the comprehensiveness and accuracy of the proposed approach.

Author: Thanks for bringing it up. The claimed “high” throughput is always a relative concept when our approach is compared with similar systems. In the paper, we claim that the SDI system can achieve high-throughput hyperspectral video recording at a low bandwidth by demonstrating 900x data throughput compared to conventional methods. As far as we know, it is a first-of-its kind of a single-pixel-based hyperspectral imager in terms of data throughput and compression ratio. Even so, the reported resolution is still far from reaching the limit of the current SDI system. It is straightforward to simultaneously increase the spatial resolution, spectral resolution, and frame rate, as discussed in Line 812-858. In a handful of compressive imaging papers, the authors sometimes perform additional interpolation to increase the resolution beyond that of the optical modulators. We have not done so in this case and quote the spatial and spectral resolution on how the micromirrors have been ganged and achieve the corresponding resolution.

In this paper, we did not aim to push the resolution limit but attempt to select resolutions appropriate for the scene while maintaining reconstruction quality. Some research works on grayscale SPI aim to push the frame rate to its limit via mechanically boosting pattern rate to ~ 10 MHz which, certainly, leads to very high frame rate but at the cost of the reconstruction and spatial resolution as a tradeoff. Some 2D array detector-based systems can achieve higher resolution in one or more dimensions but they do not enjoy the benefits of the single-pixel based approach and the high compression ratio as discussed in Line 775-811.

3. The assertion made in line 152 regarding a “900x data throughput” is not without potential ambiguities. It is important to acknowledge that utilizing a DMD-based single-pixel imaging system itself holds the capability to achieve elevated temporal or spatial resolutions. The system proposed in this paper, however, appears to trade off temporal and spatial resolution in favor of spectral resolution. The claim of “900x data throughput” warrants further scrutiny and clarification, taking into account the nuanced trade-offs between these different aspects of imaging performance.

Author: We are a little unclear on the reviewer’s comment but we would like to address it according to our understanding of “utilizing a DMD-based single-pixel imaging system itself holds the capability to achieve elevated temporal or spatial resolutions”. It is literally and naturally true that “multi-pixel” image/video formed from a ‘single-pixel’ detector has “elevated” spatial/temporal resolution. However, current single-pixel methods cannot achieve such high compression ratio of 900:1 with similar reconstruction quality for imaging hyperspectral video at similar resolution. As shown in Fig. 6 in the main paper, for a 128×128 grayscale image, a compression ratio of 10:1 can be achieved through SPI but a compression ratio of 100:1 cannot produce a normal image. This is because the data in the 2D space are far less sparse than in the 4D space. As far as we know, our paper is the first to fully explore and utilize such 4D sparsity for single-pixel based hyperspectral video imaging by proposing a new architecture and new algorithms.

As we know, simply reducing the number of measurements without optimizing the sensing and reconstruction scheme directly leads to worsened reconstruction quality. Similarly, simply boosting pattern rate for realizing high frame rate results in very low measurement SNR. Unlike such methods, we achieve high compression ratio by fully exploiting the inherent redundancy of the signal in 4D space as discussed in Line 802-811. The fundamental contribution or innovation that enables the high compression ratio in the SDI system resides in two aspects: first, the encoding scheme proposed allows the spatial-spectral information to be jointly and maximumly acquired through a small number of measurements and that the temporal correlation (optical flow) can be extracted from the same raw measurements, consequently embedding more information in each measurement; secondly, the proposed signal prior of joint sparsity in

4D space are embodied in the reconstruction algorithms, extracting more information from the compressed measurements.

Regarding resolution tradeoff, as discussed in Line 812-858, it is straightforward to simultaneously increase the resolution in all dimensions for the SDI. Besides, since signal sparsity grows when increasing the resolution of any of the spatial, spectral, or temporal dimensions, an even higher compression ratio could be achieved at higher resolutions. However, there will be a limit where signal strength compared to noise will make each measurement indistinguishable in the detector output and the A/D encoding and thus result in the inability to accurately reconstruct the signal. In addition, the temporal compression is based on the 5 kHz sampling rate and the motion of the objects in the scene. Indeed, if we tried to image faster motion, we would have to reduce the number of spatial-spectral patterns being grouped per frame as discussed in our earlier CS-MUVI publication and correspondingly may reduce the spatial/spectral pixel count to maintain the same quality.

4. The comparative advantages of this system in relation to other hyperspectral video acquisition systems are not clearly apparent in the paper's presentation. Further comprehensive discussion and substantiation are required to elucidate and establish these potential differentiators.

Author: In the *Discussion* section, we have added a detailed and thorough analysis on our approach compared to existing methods in literature. The advantages of using a single-pixel detector vs. 2D array detector are presented in Line 775-787. The motivation and novelty of the achieved high compression ratio of 900:1 are discussed in Line 787-811. The analysis on further increasing resolution is presented in Line 812-858.

5. The imaging model of the optical modulation in Fig. 1 is unclear. The imaging model is crucial for the reconstruction as the data fidelity term. Particularly, the purpose of the slit and the diffraction grating should be described and analyzed. The focal plane of each lens should be indicated.

Author: The imaging model has been described in detail in section 2.2 CS-based Joint Spatial-Spectral Encoding. Figure 2 illustrates the forward imaging model in a visualized and more straightforward way. Equation 1 formally describes the forward imaging model for sensing one frame. Equation 2 formally describes the forward imaging model for sensing multiple frames in sequence in a video. The purpose of the slit, diffraction grating, focal planes of each lens are presented Section 2.1 with edited contents.

REVIEWERS' COMMENTS

Reviewer #1 (Remarks to the Author):

The authors have comprehensively addressed my previous comments. The revised manuscript reads excellently. I recommend it for publication.

Reviewer #2 (Remarks to the Author):

The revised version of the manuscript has addressed the majority of the concerns raised in my original report. In particular, the Introduction and Discussion sections have been significantly improved with the addition of a comparison of the techniques discussed in this paper with other current work in the field. The reader will have a much better understanding of the advantages offered by the techniques discussed in this paper and appreciate the novelty of the approach. The figures and figure legends are much improved. The authors addressed typos, grammar issues, etc. as suggested in my original report. I still think the structure of the paper + supplement is awkward. However, I recommend the publication of the paper in its current form.

Reviewer #3 (Remarks to the Author):

The novelty and contribution should be further emphasized to meet the bar of NC. It is vital to demonstrate this paper is not a trivial combination of existing methods/techniques. NC would not accept papers without clear and strong novelty and contributions.

Response to Reviewers' comments

Reviewer #1 (Remarks to the Author):

The authors have comprehensively addressed my previous comments. The revised manuscript reads excellently. I recommend it for publication.

Author: Thank you for help reviewing the manuscript and for your valuable suggestions.

Reviewer #2 (Remarks to the Author):

The revised version of the manuscript has addressed the majority of the concerns raised in my original report. In particular, the Introduction and Discussion sections have been significantly improved with the addition of a comparison of the techniques discussed in this paper with other current work in the field. The reader will have a much better understanding of the advantages offered by the techniques discussed in this paper and appreciate the novelty of the approach. The figures and figure legends are much improved. The authors addressed typos, grammar issues, etc. as suggested in my original report. I still think the structure of the paper + supplement is awkward. However, I recommend the publication of the paper in its current form.

Author: Thank you for help reviewing the manuscript and for your valuable suggestions.

Reviewer #3 (Remarks to the Author):

The novelty and contribution should be further emphasized to meet the bar of NC. It is vital to demonstrate this paper is not a trivial combination of existing methods/techniques. NC would not accept papers without clear and strong novelty and contributions.

Author: Thank you for your comments. The fundamental contribution or innovation that enables the high compression ratio or data throughput in the SDI system is the proposed framework which fully exploits signal sparsity in 4D space in both sensing and recovery: the encoding scheme proposed allows the spatial-spectral information to be jointly and maximally acquired through a small number of measurements and the temporal correlation (optical flow) can be extracted from the same raw measurements, consequently embedding more scene information in each measurement; the signal prior of joint sparsity in 4D space are proposed and realized in the reconstruction, extracting more information from the compressed measurements.